# Robust Fine-tuning of Zero-shot Models via Variance Reduction

**Beier Zhu**   **Jiequan Cui**   **Hanwang Zhang**
Nanyang Technological University
`beier002@e.ntu.edu.sg, hanwangzhang@ntu.edu.sg`

## Abstract

When fine-tuning zero-shot models like CLIP, our desideratum is for the fine-tuned model to excel in both in-distribution (ID) and out-of-distribution (OOD). Recently, ensemble-based models (ESM) have been shown to offer significant robustness improvement, while preserving high ID accuracy. However, our study finds that ESMs do not solve the ID-OOD trade-offs: they achieve peak performance for ID and OOD accuracy at different mixing coefficients. When optimized for OOD accuracy, the ensemble model exhibits a noticeable decline in ID accuracy, and vice versa. In contrast, we propose a sample-wise ensembling technique that can simultaneously attain the best ID and OOD accuracy without the trade-offs. Specifically, we construct a Zero-Shot Failure (ZSF) set containing training samples incorrectly predicted by the zero-shot model. For each test sample, we calculate its distance to the ZSF set and assign a higher weight to the fine-tuned model in the ensemble if the distance is small. We term our method Variance Reduction Fine-tuning (VRF), as it effectively reduces the variance in ensemble predictions, thereby decreasing residual error. On ImageNet and five derived distribution shifts, our VRF further improves the OOD accuracy by 1.5 - 2.0 pp over the ensemble baselines while maintaining or increasing ID accuracy. VRF achieves similar large robustness gains (0.9 - 3.1 pp) on other distribution shifts benchmarks. Codes are available in `https://github.com/BeierZhu/VRF`.

## 1   Introduction

To ensure the reliability of machine learning systems, it is essential to develop models that can generalize to unseen, out-of-distribution environments. Large pre-trained models such as CLIP [20] and ALIGN [10] have recently shown remarkable robustness against challenging distribution shifts. However, it is widely acknowledged that these improvements in robustness are most pronounced in the zero-shot setting, while conventional fine-tuning on these models often compromises robustness when compared to zero-shot performance [28, 15, 14]. This phenomenon is known as the ID-OOD trade-offs, *i.e.*, improving performance on in-distribution (ID) data can sometimes lead to decreased performance on out-of-distribution (OOD) data [12, 25].

In recent years, ensemble-based models (ESMs) have demonstrated significant success in addressing the ID-OOD dilemma [17, 28, 14, 31]. Specifically, denote the input as $\mathbf{x}$, the zero-shot model as $\hat{\mathbb{P}}(y|\mathbf{x}; \theta_{\mathsf{zs}})$ and the fine-tuned model as $\hat{\mathbb{P}}(y|\mathbf{x}; \theta_{\mathsf{ft}})$, existing ESMs typically employ the output-space ensemble (OSE) [14, 31], which outputs $\hat{\mathbb{P}}(y|\mathbf{x}; \theta_{\mathsf{ose}}) = \alpha\hat{\mathbb{P}}(y|\mathbf{x}; \theta_{\mathsf{ft}}) + (1 - \alpha)\hat{\mathbb{P}}(y|\mathbf{x}; \theta_{\mathsf{zs}})$, and the weight-space ensemble (WSE) [28, 17], which outputs $\hat{\mathbb{P}}(y|\mathbf{x}; \theta_{\mathsf{wse}}) = \hat{\mathbb{P}}(y|\mathbf{x}; \alpha\theta_{\mathsf{ft}} + (1 - \alpha)\theta_{\mathsf{zs}})$, where $\alpha \in [0, 1]$. Compared to fine-tuned models, ESMs offer significant accuracy enhancements under distribution shift, while maintaining high ID accuracy.

However, ESM cannot fully address the ID-OOD trade-offs. In Figure 1 (a), by varying the mixing coefficient $\alpha$, we plot the ID-OOD frontier curves (pink line) for the CLIP ViT-B/16 model on

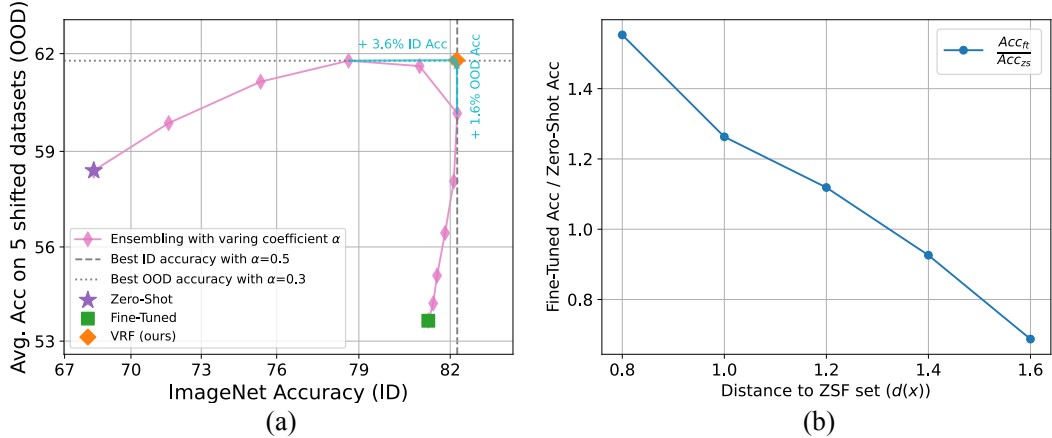

(a)                  (b)

Figure 1: (a) ID-OOD frontier curves for the CLIP ViT-B/16 model on the ID (ImageNet) and OOD (IN-{V2, R, A, Sketch} and ObjectNet) datasets by varying the mixing coefficient $\alpha$. The ensemble model achieves its best ID and OOD performance at different $\alpha$ values. Our method VRF simultaneously attains the best ID and OOD accuracy, outperforming the ensemble by $3.6\%$ on OOD and $1.6\%$ on ID at its optimal performance points.(b) Relationship between the ratio of fine-tuned accuracy to zero-shot accuracy ($\frac{\text{Acc}_{\text{ft}}}{\text{Acc}_{\text{zs}}}$) and the distance to the zero-shot failure set ($d(\mathbf{x})$). $\frac{\text{Acc}_{\text{ft}}}{\text{Acc}_{\text{zs}}}$ demonstrates a monotonic decrease as $d(\mathbf{x})$ increases.

ImageNet [3] (ID) and five derived distribution-shifted datasets (OOD): ImageNet-V2 [21], ImageNet-R [7], ImageNet-A [9], ImageNet-Sketch [27] and ObjectNet [1]. We find that the ensemble model achieves its optimal ID and OOD performance at different $\alpha$ values: the best ID accuracy is achieved at $\alpha = 0.5$ and the best OOD accuracy is obtained at $\alpha = 0.3$. When the ensemble model reaches its optimal value for OOD, the performance on ID decreases by $3.6\%$ relative to its peak. Similarly, when the ensemble model is optimized for ID, the performance on OOD decreases by $1.6\%$ relative to its best value – the ID-OOD trade-offs still persist for ESMs. This raises a natural question:

*Can ensemble-based models simultaneously attain the best ID and OOD accuracy?*

In this paper, we affirmatively answer this question by proposing a sample-wise ensembling technique, dubbed variance reduction fine-tuning (VRF). This method is motivated by an empirical finding illustrated in Fig 1 (b). For each sample in the training dataset, if the fine-tuned model correctly predicts the label while the zero-shot model fails, we collect its features representation in the fine-tuned model as the zero-shot failure (ZSF) set. We then measure the distance $d(\mathbf{x})$ of each test sample $\mathbf{x}$ to the ZSF set. Based on this distance, test samples are grouped into bins, and we compute the ratio of fine-tuned accuracy to zero-shot accuracy: $\frac{\text{Acc}_{\text{ft}}}{\text{Acc}_{\text{zs}}}$ for each bin (implementation details are in Section C.7). Interestingly, we observe that the ratio $\frac{\text{Acc}_{\text{ft}}}{\text{Acc}_{\text{zs}}}$ monotonically decreases as $d(\mathbf{x})$ increases. Intuitively, the closer a sample is to the ZSF set, the more likely it is that the zero-shot model makes incorrect predictions, whereas the fine-tuned model is more likely to be accurate, leading to a higher $\frac{\text{Acc}_{\text{ft}}}{\text{Acc}_{\text{zs}}}$ ratio. Therefore, we use the distance to assign weights to the models: a smaller $d(\mathbf{x})$ results in a higher weight for the fine-tuned model, and vice versa.

As depicted by the orange diamond in Fig. 1 (a), by leveraging the sample-wise weights, our VRF simultaneously attains the best ID and OOD accuracy. In Section 5, we show that on a variety of different models and tasks, our VRF approach consistently outperforms the existing fine-tuning and ensembling methods, including linear probing, end-to-end fine-tuning, LP-FT [15], OSE and WSE [28]. In specific, on ImageNet and five derived distribution shifts, our VRF further improves the OOD accuracy by 1.5 - 2.0 pp over the ensemble baselines while maintaining or increasing ID accuracy. Furthermore, in Section 4, we justify our approach by demonstrating that it effectively minimizes the variance of the ensemble models, resulting in reduced residual error.

## 2 Related Work

**Mitigating ID-OOD trade-offs.** Improving performance on in-distribution data can sometimes lead to a decrease in performance on out-of-distribution data, and vice versa. This phenomenon is known as the ID-OOD trade-offs. Xie et al. [29] leverage auxiliary information as outputs of auxiliary tasks to pre-train a model to reduce OOD error. Khani and Liang [12] show that self-training on large amounts of unlabeled data can mitigate such trade-offs by removing spurious features. Tripuraneni et al. [25] tackle this problem by learning representations that are robust across diverse tasks. However, these methods usually necessitate additional unlabeled data or auxiliary information. In contrast, our VRF is a straightforward variation of fine-tuning that does not require any extra data.

**Robust fine-tuning of zero-shot models.** Vision-language models like CLIP [20] have demonstrated outstanding improvements in robustness. It is commonly acknowledged that conventional fine-tuning methods often compromise robustness when compared to zero-shot performance. Therefore, enhancing downstream robustness has been the focus of subsequent works [15, 28, 5, 19, 6, 30]. Kumar et al. [15] show that a two-process of linear probing followed by full fine-tuning can alleviate feature distortion, leading to stronger OOD performance without sacrificing ID accuracy. Wortsman et al. [28] propose a method of weight interpolation between the zero-shot and the fine-tuned models to improve both ID and OOD accuracy. Goyal et al. [5] demonstrate that mimicking the contrastive pre-training objectives to fine-tune the zero-shot models outperforms tuning via the traditional supervised cross-entropy loss. However, the ID-OOD trade-offs are still observed with these methods. In contrast, our method VRF can simultaneously achieve the best ID and OOD accuracy.

## 3 Methods

### 3.1 Set Up

**Task:** Consider a classification setting where the goal is to map an instance $\mathbf{x} \in \mathcal{X}$ to a label $y \in \mathcal{Y} = [K]$. We are provided with a zero-shot model $f(\cdot; \theta_{\mathsf{zs}})$, a downstream dataset $\mathcal{D} = \{\mathbf{x}_i, y_i\}_{i=1}^N$, and a fine-tuned model $f(\cdot; \theta_{\mathsf{ft}})$ which is trained on $\mathcal{D}$. Below, we outline the implementation of the zero-shot and fine-tuned models:

- **Zero-shot models** (ZS): We investigate CLIP models [20] as our zero-shot models. CLIP models are pre-trained using image-text pairs $\{(\mathbf{x}_1, \mathbf{t}_1), ..., (\mathbf{x}_B, \mathbf{t}_B)\}$ from the Internet. The objective of the CLIP models is to train a visual encoder $\Phi_{\mathsf{v}}$ and a text encoder $\Phi_{\mathsf{t}}$ such that the cosine similarity $< \Phi_{\mathsf{v}}(\mathbf{x}_i), \Phi_{\mathsf{t}}(\mathbf{t}_i) >$ is maximized relative to unmatched pairs. CLIP models perform zero-shot inference for $K$ classes by matching $\mathbf{x}$ with potential class names $\{c_1, ..., c_K\}$. Concretely, by extending the class name $\{c_k\}$ to a prompt "$\mathbf{t}_k =$ a photo of a $\{c_k\}$", the zero-shot model outputs the score (logit) for class $k$ as $f(\mathbf{x}; \theta_{\mathsf{zs}})_k = < \Phi_{\mathsf{v}}(\mathbf{x}), \Phi_{\mathsf{t}}(\mathbf{t}_k) >$. The predicted probabilities can be calculated using the softmax function, *i.e.*, $\hat{\mathbb{P}}(y|\mathbf{x}; \theta_{\mathsf{zs}}) = \mathrm{softmax}(f(\mathbf{x}; \theta_{\mathsf{zs}}))_y$. The model outputs the label as $\mathrm{pred}(f(\mathbf{x}; \theta_{\mathsf{zs}})) = \mathrm{argmax}_i f(\mathbf{x}; \theta_{\mathsf{zs}})_i$

- **Linear classifiers** (LC): We learn a linear classifier on top of the visual embedding $\Phi_{\mathsf{v}}(\mathbf{x})$ while freezing the visual encoder $\Phi_{\mathsf{v}}$. The parameters of the linear classifier are optimized to minimize the cross-entropy loss on $\mathcal{D}$.

- **End-to-end fine-tuning** (E2E-FT): We update both the linear classifier and the visual encoder by minimizing the cross-entropy loss on $\mathcal{D}$.

- **Linear probing then full fine-tuning** [15] (LP-FT): We employ a two-phase fine-tuning approach: initially training a linear classifier, followed by full fine-tuning starting from the solution derived from training the linear classifier.

- **Output-space ensemble** (OSE): We perform linear interpolation of the outputs between a zero-shot model and a fine-tuned model (*e.g.*, E2E-FT or LP-FT):

$$\hat{\mathbb{P}}(y|\mathbf{x}; \theta_{\mathsf{ose}}) = \alpha\hat{\mathbb{P}}(y|\mathbf{x}; \theta_{\mathsf{ft}}) + (1 - \alpha)\hat{\mathbb{P}}(y|\mathbf{x}; \theta_{\mathsf{zs}}), \text{ where } \alpha \in [0, 1] \tag{1}$$

- **Weight-space ensemble** [28] (WSE). We combine the weights through linear interpolation between a zero-shot model and a fine-tuned model:

$$\hat{\mathbb{P}}(y|\mathbf{x}; \theta_{\mathsf{wse}}) = \hat{\mathbb{P}}(y|\mathbf{x}; \alpha\theta_{\mathsf{ft}} + (1 - \alpha)\theta_{\mathsf{zs}}), \text{ where } \alpha \in [0, 1] \tag{2}$$

---

**Algorithm 1** Variation Reduction Fine-tuning

---

1: **Given**: Training dataset $\mathcal{D}$, a zero-shot model $f_{\mathsf{zs}}$ and a fine-tuned model $f_{\mathsf{ft}}$.
2: Build zero-shot failure set $\mathcal{V}$ using Eq. (3).                    ▷ Step 1: Identification
3: **Inference Stage:**
4: Given a test sample $\mathbf{x}$, compute its feature representation $\mathbf{v}$, zero-shot prediction $\hat{\mathbb{P}}_{\mathsf{zs}}(y|\mathbf{x})$ and fine-tuned model prediction $\hat{\mathbb{P}}_{\mathsf{ft}}(y|\mathbf{x})$.
5: Compute the $k$-NN distance to $\mathcal{V}$ as $d(\mathbf{x})$ using Eq. (4).        ▷ Step 2: Distance Calculation
6: Compute the weight $\omega(\mathbf{x})$ using Eq. (6).
7: Return $\hat{\mathbb{P}}_{\mathsf{vrf}}(y|\mathbf{x})$ using Eq. (5)                    ▷ Step 3: Sample-Wise Ensembling

---

## 3.2 Variance Reduction Fine-tuning

We now present our proposed method, VRF, which consists of three steps. First, before the inference stage, we collect the Zero-Shot Failure (ZSF) set. Second, for a given test sample, we calculate its distance to the ZSF set. Third, we assign weights to combine predictions from the zero-shot and fine-tuned models based on this distance.

**Step 1 (Identification).** For each $\mathbf{x}_i$ in the training dataset $\mathcal{D}$, if the fine-tuned model correctly predicts the label while the zero-shot model fails, we collect its feature representation $\mathbf{v}_i = \Phi_{\mathsf{v}}(\mathbf{x}_i; \theta_{\mathsf{ft}})$ from the fine-tuned model to form the zero-shot failure set $\mathcal{V}$. Specifically, $\mathcal{V}$ is defined as:

$$\mathcal{V} = \{\mathbf{v}_i \text{ s.t. } y_i = \mathrm{pred}(f_{\mathsf{ft}}(\mathbf{x}_i)) \text{ and } y_i \neq \mathrm{pred}(f_{\mathsf{zs}}(\mathbf{x}_i))\}. \tag{3}$$

Here, $f_{\mathsf{zs}}(\cdot)$ and $f_{\mathsf{ft}}(\cdot)$ are used to denote $f(\cdot; \theta_{\mathsf{zs}})$ and $f(\cdot; \theta_{\mathsf{ft}})$, respectively, for simplicity.

**Step 2 (Distance Calculation).** The key empirical observation underpinning VRF is that in the vicinity of the ZSF set, a test sample typically exhibits lower zero-shot accuracy ($\mathrm{Acc}_{\mathsf{zs}}$) and higher fine-tuned accuracy ($\mathrm{Acc}_{\mathsf{ft}}$). Consequently, the $\frac{\mathrm{Acc}_{\mathsf{ft}}}{\mathrm{Acc}_{\mathsf{zs}}}$ ratio demonstrates a monotonic decrease as the distance from the sample to the ZSF set increases. In this paper, we adopt non-parametric density estimation using nearest neighbors [24] to measure the distance of a test sample to the dataset $\mathcal{V}$. Specifically, during inference, we derive the feature representation $\mathbf{v}$ of a test sample $\mathbf{x}$, and compute the $\ell_2$ distances $\|\mathbf{v} - \mathbf{v}_i\|_2$ w.r.t. $\mathbf{v}_i \in \mathcal{V}$. We reorder $\mathcal{V}$ according to the increasing $\ell_2$ distance and denote the ordered set in sequence as $\mathcal{V}' = (\mathbf{v}_{(1)}, \mathbf{v}_{(2)}, ..., \mathbf{v}_{(|\mathcal{V}|)})$. The distance of $\mathbf{x}$ to $\mathcal{V}$ is defined as the $\ell_2$ distance to the $k$-th nearest neighbor ($k$-NN), *i.e.*,

$$d(\mathbf{x}; \mathcal{V}, k) = \|\mathbf{v} - \mathbf{v}_{(k)}\|_2. \tag{4}$$

If there is no ambiguity, we use $d(\mathbf{x})$ to denote $d(\mathbf{x}; \mathcal{V}, k)$ for readability. Since the features in CLIP models are $\ell_2$ normalized, $d(\mathbf{x})$ are bounded between $[0, 2]$.

**Step 3 (Sample-Wise Ensembling).** We implement sample-wise out-space ensembling in the form:

$$\hat{\mathbb{P}}_{\mathsf{vrf}}(y|\mathbf{x}) = \omega(\mathbf{x}) \cdot \hat{\mathbb{P}}_{\mathsf{ft}}(y|\mathbf{x}) + (1 - \omega(\mathbf{x})) \cdot \hat{\mathbb{P}}_{\mathsf{zs}}(y|\mathbf{x}), \tag{5}$$

where $\omega(\mathbf{x}) \in (0, 1)$. We use the distance to ZSF set $d(\mathbf{x})$ to determine the weight $\omega$. As shown by the blue line in Fig 2, a smaller value of $d(\mathbf{x})$ corresponds to a larger $\frac{\mathrm{Acc}_{\mathsf{ft}}}{\mathrm{Acc}_{\mathsf{zs}}}$ ratio, and vice versa. Therefore, we set the weight $\omega$ to be inversely proportional to $d(\mathbf{x})$. Given that $\omega$ is bounded between 0 and 1, we employ a sigmoid function $\sigma(\cdot)$ as:

$$\omega(\mathbf{x}) = \sigma(-(d(\mathbf{x}) - a)/b), \tag{6}$$

where $a, b > 0$ are two hyper-parameters swept using

Figure 2: Relationship between $\frac{\mathrm{Acc}_{\mathsf{ft}}}{\mathrm{Acc}_{\mathsf{zs}}}$ and the weight $\omega(\mathbf{x})$.

accuracy on ID validation set. We visualize the weight curve in green on Fig 2, setting $a = 1.5$ and $b = 0.6$. We summarize the whole process in Algorithm 1.

# 4 Justification

We now prove that our VRF can effectively reduce the variance of the combined model, resulting in lower errors compared to ensembling using a constant mixing coefficient.

### 4.1 Background

The outputs of a well trained classifier are expected to approximate the *a posterior* class distribution. Apart from the irreducible error (Bayes error), the residual error of a classifier can be broken down into bias and variance components. In specific, for a test sample $\mathbf{x}$, the probability output of a classifier parameterized by $\theta$ can be expressed as:

$$\hat{\mathbb{P}}(y|\mathbf{x};\theta) = \mathbb{P}(y|\mathbf{x}) + \underbrace{\beta_y + \eta_y(\mathbf{x})}_{\text{residual error for } \mathbf{x}}, \tag{7}$$

where $\mathbb{P}(y|\mathbf{x})$ denotes the true *a posterior* distribution, $\beta_y$ is the label bias of $\hat{\mathbb{P}}(y|\mathbf{x};\theta)$ which is independent to the input $\mathbf{x}$, and $\eta_y(\mathbf{x})$ is related to the given input $\mathbf{x}$. In this study, we primarily attribute the residual error to the variance term (*i.e.*, $\beta_y = 0$), as the label bias problem in foundation models has been effectively addressed by Zhu et al. [31]. Tumer et al. [26] have proven that the expected residual error $E$ is given by:

$$E = \frac{\mathbb{V}[\eta_y(\mathbf{x})]}{s}, \tag{8}$$

where $s$ is a constant factor related to the derivative of the true *a posterior* distribution and is independent of the trained model, and $\mathbb{V}[\eta_y(\mathbf{x})]$ is the variance.

### 4.2 Variance Reduction Fine-tuning Leads to Lower Residual Error

Let us shift our focus to the effects of combining the zero-shot and fine-tuned models. Let $g_{\mathsf{zs}}(\cdot)$ and $g_{\mathsf{ft}}(\cdot)$ be two functions that produce weights for ensembling the models. Subject to the constraint that $g_{\mathsf{zs}}(\mathbf{x}) + g_{\mathsf{ft}}(\mathbf{x}) = 1$, the residual error of the combined classifier is obtained by:

$$\hat{\mathbb{P}}_{\mathsf{vrf}}(y|\mathbf{x}) = g_{\mathsf{zs}}(\mathbf{x})\hat{\mathbb{P}}_{\mathsf{zs}}(y|\mathbf{x}) + g_{\mathsf{ft}}(\mathbf{x})\hat{\mathbb{P}}_{\mathsf{ft}}(y|\mathbf{x}) = \mathbb{P}(y|\mathbf{x}) + \underbrace{g_{\mathsf{zs}}(\mathbf{x}) \cdot \eta_{\mathsf{zs}}(\mathbf{x}) + g_{\mathsf{ft}}(\mathbf{x}) \cdot \eta_{\mathsf{ft}}(\mathbf{x})}_{\eta_{\mathsf{vrf}}(\mathbf{x})}, \tag{9}$$

where we omit the subscript $y$ of $\eta$ for readability. The variance of $\eta_{\mathsf{vrf}}(\mathbf{x})$ can be expressed as:

$$\mathbb{V}[\eta_{\mathsf{vrf}}(\mathbf{x})] = g_{\mathsf{zs}}(\mathbf{x})^2 \cdot \mathbb{V}[\eta_{\mathsf{zs}}(\mathbf{x})] + g_{\mathsf{ft}}(\mathbf{x})^2 \cdot \mathbb{V}[\eta_{\mathsf{ft}}(\mathbf{x})]. \tag{10}$$

Here, we assume the residual errors are independent following the assumption of the previous studies of CLIP fine-tuning [14, 31]. We further explore the case of correlated residual errors in Section B. According to Eq. (8), the reduction in variance can be readily translated into a reduction in error rates. To obtain the smallest variance $\mathbb{V}[\eta_{\mathsf{vrf}}(\mathbf{x})]$, we minimize Eq. (10) using Lagrange multiplier to enforce the constraint that $g_{\mathsf{zs}}(\mathbf{x}) + g_{\mathsf{ft}}(\mathbf{x}) = 1$, and obtain the optimal weight function $g_{\mathsf{ft}}$ as:

$$g_{\mathsf{ft}}(\mathbf{x}) = \frac{\mathbb{V}[\eta_{\mathsf{zs}}(\mathbf{x})]}{\mathbb{V}[\eta_{\mathsf{zs}}(\mathbf{x})] + \mathbb{V}[\eta_{\mathsf{ft}}(\mathbf{x})]} = \frac{E_{\mathsf{zs}}}{E_{\mathsf{zs}} + E_{\mathsf{ft}}} = (1 + \frac{E_{\mathsf{ft}}}{E_{\mathsf{zs}}})^{-1} \propto \frac{\text{Acc}_{\mathsf{ft}}}{\text{Acc}_{\mathsf{zs}}} \tag{11}$$

Since $\frac{\text{Acc}_{\mathsf{ft}}}{\text{Acc}_{\mathsf{zs}}} \propto d(\mathbf{x})^{-1}$ (a smaller distance $d(\mathbf{x})$ is associated with a larger $\frac{\text{Acc}_{\mathsf{ft}}}{\text{Acc}_{\mathsf{zs}}}$ as shown in Fig. 2), we design the weighting function $g_{\mathsf{ft}}(\mathbf{x}) = \omega(\mathbf{x}) \propto d(\mathbf{x})^{-1}$ as in Eq. (6).

## 5 Experiments

### 5.1 Experimental Setup

**Datasets with distribution shifts.** We provide the results for ImageNet [3] and its five derived distribution shifts: (1) ImageNet-V2 (IN-V2) [21]: Test images sampled after a decade of the original ImageNet. (2) ImageNet-R (IN-R) [7]: Contains renditions (*e.g.*, art, cartoons, graffiti). (3) ImageNet Sketch (IN-Sketch) [27]: Consists of sketches rather than natural photos. (4) ImageNet-A (IN-A) [9]: Collects real-world images that are misclassified by ResNet models. (5) ObjectNet [1], a test set featuring objects with diverse backgrounds, rotations, and imaging viewpoints. We extend our analysis to include a standard distribution shift benchmark [15, 14, 4]: CIFAR-10 $\rightarrow$ STL-10, where the ID is CIFAR-10 [13], and the OOD is STL-10 [2]. We removed the "monkey" class from STL-10, as it does not exist in CIFAR-10. In addition, we also consider subpopulation shifts, where the ID data contains a few sub-categories, and the OOD data comprises different sub-categories within the

Table 1: Accuracy of various methods on ImageNet and derived distribution shifts for CLIP ViT-B/32.

| Method | IN | Distribution shifts | | | | | Avg shifts |
| --- | --- | --- | --- | --- | --- | --- | --- |
| | | IN-V2 | IN-Sketch | IN-A | IN-R | ObjectNet | |
| Zero-shot [20] | 63.3 | 55.9 | 42.3 | 31.5 | 69.3 | 43.5 | 48.5 |
| Linear classifier [20] | 75.4 | 63.4 | 38.8 | 26.1 | 58.7 | 41.5 | 45.7 |
| E2E-FT [28] | 76.2 | 64.2 | 38.7 | 21.0 | 57.1 | 40.1 | 44.2 |
| + Weight-space ensemble [28] | 77.9 | 67.2 | 45.1 | 28.8 | 66.4 | 45.1 | 50.5 |
| + Output-space ensemble | 77.3 | 66.0 | 44.2 | 27.1 | 68.4 | 44.4 | 50.0 |
| + VRF (ours) | 77.6 | 66.7 | 47.0 | 29.2 | 70.9 | 46.3 | 52.0 |
| Δ | +0.3 | +0.7 | +2.8 | +2.1 | +2.5 | +1.9 | +2.0 |
| LP-FT [15] | 76.9 | 64.8 | 39.9 | 25.7 | 69.9 | 42.6 | 48.6 |
| + Weight-space Ensemble [28] | 78.0 | 67.0 | 44.8 | 31.2 | 65.8 | 46.1 | 51.0 |
| + Output-space Ensemble | 77.8 | 66.3 | 44.0 | 29.5 | 66.2 | 45.5 | 50.3 |
| + VRF (ours) | 77.8 | 66.7 | 46.1 | 31.0 | 70.0 | 46.3 | 51.8 |
| Δ | +0.0 | +0.4 | +2.1 | +1.5 | +3.8 | +0.8 | +1.5 |

Table 2: Accuracy of various methods on ImageNet and derived distribution shifts for CLIP ViT-B/16.

| Method | IN | Distribution shifts | | | | | Avg shifts |
| --- | --- | --- | --- | --- | --- | --- | --- |
| | | IN-V2 | IN-Sketch | IN-A | IN-R | ObjectNet | |
| Zero-shot [20] | 68.3 | 61.9 | 48.3 | 50.1 | 77.6 | 54.2 | 58.4 |
| Linear classifier [20] | 79.3 | 69.1 | 44.8 | 44.3 | 66.7 | 51.1 | 55.2 |
| E2E-FT [28] | 81.3 | 70.6 | 45.1 | 36.6 | 65.6 | 50.5 | 53.7 |
| + Weight-space ensemble [28] | 82.5 | 73.1 | 51.6 | 47.6 | 75.1 | 55.7 | 60.6 |
| + Output-space ensemble | 82.2 | 72.0 | 50.6 | 46.8 | 76.7 | 54.9 | 60.2 |
| + VRF (ours) | 82.3 | 72.1 | 52.9 | 48.4 | 78.7 | 56.4 | 61.8 |
| Δ | +0.1 | +0.1 | +2.3 | +1.6 | +2.0 | +1.5 | +1.6 |
| LP-FT [15] | 81.5 | 70.7 | 46.7 | 41.4 | 66.4 | 52.4 | 55.5 |
| + Weight-space ensemble [28] | 82.4 | 73.0 | 51.5 | 50.6 | 74.2 | 56.6 | 61.2 |
| + Output-space ensemble | 82.1 | 72.3 | 50.9 | 50.9 | 74.9 | 55.7 | 60.9 |
| + VRF (ours) | 82.1 | 72.3 | 52.9 | 51.2 | 78.8 | 57.2 | 62.4 |
| Δ | +0.0 | +0.0 | +2.0 | +0.3 | +3.9 | +1.5 | +1.5 |

same parent category. Following [15, 14], we adopt Entity30 dataset [23], which aims to categorize images into one of 30 entity categories, such as "vehicle" and "insect".

**Baselines.** We adopt two models: CLIP ViT-B/32 and a larger ViT-B/16 from OpenAI [20]. The default model used in ablation studies is the CLIP ViT-B/16. In addition to the zero-shot models, we compare our approach against five standard methods for adapting pre-trained models: (1) linear classifier [20], (2) E2E-FT, (3) LP-FT [15], (4) OSE, and (5) WSE [28]. The descriptions of these methods have been included in Section 3.1.

**Implementation details.** When fine-tuning E2E-FT models, we adhere to Wortsman et al. [28], employing the default PyTorch AdamW optimizer for 10 epochs with weight decay of 0.1 and a cosine-annealing learning rate schedule with 500 warm-up steps. Unless specified, we use a learning rate of $3 \times 10^{-5}$, gradient clipping at norm 1. When fine-tuning LP-FT, we first adopt the settings of Wortsman et al. [28] to train the linear classifier, then full fine-tune the models at a learning rate of $1 \times 10^{-5}$. For efficiently performing $k$-NN search, we use Faiss library [11]. Denote the size of the ZSF set is $|\mathcal{V}|$, we scale the $k$ according to a percentage $p\%$ of the sample set, where $k = \text{floor}(p\% \cdot |\mathcal{V}|)$. In this paper, $p$ is set to $0.1\%$, a value consistent with the default setting proposed by Sun et al. [24]. Note that all the hyperparameters, *e.g.*, $\alpha, a, b$, are searched using the accuracy on the in-distribution (ID) validation set. Derived distribution shift datasets are *only for evaluation and not for hyperparameter sweeps*. See Appendix C.1 for the details of experimental details.

Table 3: Accuracy of various methods on CIFAR-10 → STL-10 and Entity-30.

| Method | CIFAR → STL ID | CIFAR → STL OOD | Entity-30 ID | Entity-30 OOD | Method | CIFAR → STL ID | CIFAR → STL OOD | Entity-30 ID | Entity-30 OOD |
|---|---|---|---|---|---|---|---|---|---|
| Zero-shot [20] | 88.3 | 97.1 | 65.2 | 66.5 | Zero-shot [20] | 90.1 | 98.4 | 68.3 | 68.2 |
| Linear classifier | 95.0 | 96.6 | 93.3 | 68.1 | Linear classifier | 95.8 | 97.7 | 95.3 | 69.6 |
| E2E-FT [28] | 97.9 | 93.5 | 94.4 | 65.1 | E2E-FT [28] | 98.6 | 96.1 | 96.9 | 68.2 |
| + WSE [28] | 98.2 | 95.7 | 94.6 | 68.8 | + WSE [28] | 98.7 | 97.8 | 97.2 | 71.9 |
| + OSE | 97.9 | 95.9 | 94.4 | 66.4 | + OSE | 98.6 | 96.6 | 97.0 | 71.5 |
| + VRF (ours) | 97.8 | 97.3 | 94.5 | 69.5 | + VRF (ours) | 98.6 | 98.4 | 97.0 | 72.7 |
| Δ | -0.1 | +1.4 | +0.1 | +3.1 | Δ | +0.0 | +1.8 | +0.0 | +1.2 |
| LP-FT [15] | 97.9 | 95.0 | 94.6 | 67.7 | LP-FT [15] | 98.5 | 96.3 | 96.9 | 68.8 |
| + WSE [28] | 98.1 | 96.4 | 94.8 | 68.8 | + WSE [28] | 98.7 | 97.9 | 97.3 | 72.1 |
| + OSE | 98.1 | 96.4 | 94.7 | 68.5 | + OSE | 98.6 | 97.7 | 97.2 | 71.8 |
| + VRF (ours) | 98.1 | 97.5 | 94.8 | 70.1 | + VRF (ours) | 98.6 | 98.6 | 97.4 | 72.9 |
| Δ | +0.0 | +1.1 | +0.1 | +1.6 | Δ | +0.0 | +0.9 | +0.2 | +1.1 |

(a) CLIP ViT-B/32          (b) CLIP ViT-B/16

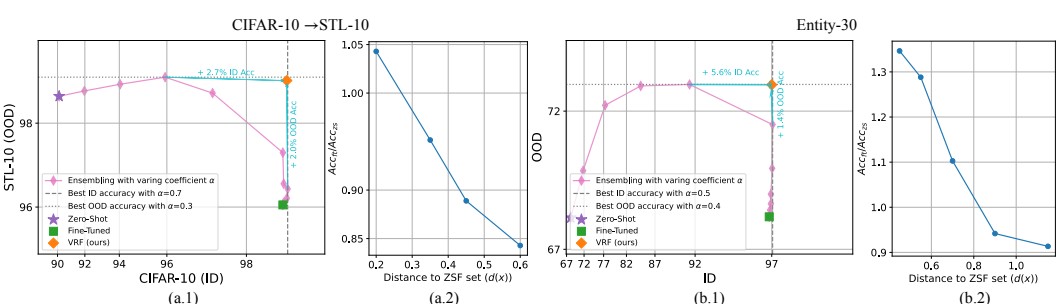

Figure 3: ID-OOD frontier curves by varying the mixing coefficient $\alpha$ and $\frac{\text{Acc}_{\text{ft}}}{\text{Acc}_{\text{zs}}}$ curves for the CLIP ViT-B/16 . (a) CIFAR-10 (ID) and STL-10 (OOD) results. (b) Entity-30 results.

## 5.2 Results

**ImageNet and its five shifted distribution results.** In Table 1 and 2, we report the ID-OOD accuracies of fine-tuning baselines for CLIP ViT-32 and CLIP ViT-16 models, respectively. For OSE and WSE, we choose the mixing coefficient $\alpha$ with the highest ID validation accuracy. To enhance clarity in the results, we denote the improvement over OSE as $\Delta$ in Tables 1 and 2. We observe that our VRF boosts the accuracy of fine-tuned models, including ensembling baseline models, across five ImageNet distribution shifted datasets, while maintaining or improving the ImageNet in-distribution performance. For instance, in Table 1, when ensembling with the E2E-FT model, our VRF outperforms the OSE model by $2.0\%$ on distribution shifts while increasing the ID accuracy by $0.3\%$. Compared to WSE models, our VRF achieves a delta of $1.2\%$ on distribution shifts, while maintaining ID performance within $0.2\%$, as shown in E2E-FT part of Table 2.

**CIFAR-10 → STL-10 and Entity-30 results.** We report the accuracy of various methods in Table 3 (a,b). We note that fine-tuning baselines can enhance the accuracy on CIFAR-10 compared to the zero-shot models. However, this improvement comes at the expense of reduced accuracy on STL-10. For instance, E2E-FT leads to a decrease of approximately $3.6\%$ in STL-10 accuracy, as shown in Table 3(a). Previous ensemble methods can mitigate the degradation to some extent, but the STL-10 performance still lags behind the zero-shot performance, *e.g.*, In Table 3(b), the accuracy of E2E-FT + WSE is $97.8\%$ whereas the zero-shot performance is $98.4\%$. In contrast, our VRF simultaneously improves accuracy on both CIFAR-10 and STL-10. Similarly, for Entity-30, our VRF can further improvement the OOD performance when compared to WSE and OSE methods.

In addition, we plot the ID-OOD frontier curves in Figure 3 (a.1&b.1), respectively. Similar to the results on ImageNet (Figure 1(a)), the ensemble model achieves its best ID and OOD performances at different $\alpha$ values. For instance, on the CIFAR-10 benchmark, when the ensemble model attains its optimal ID value at $\alpha = 0.7$, the OOD performance decreases by $2.0\%$ relative to its peak.

Table 4: Results of VRF for linear-probed models using CLIP ViT-B/16 models.

| Method | ImageNet | | CIFAR-10 | | Entity-30 | |
|---|---|---|---|---|---|---|
| | ID | OOD | ID | OOD | ID | OOD |
| Zero-shot classifier [20] | 68.3 | 58.4 | 90.1 | 98.4 | 68.3 | 68.2 |
| Linear classifier | 79.3 | 55.2 | 95.8 | 97.7 | 95.3 | 69.6 |
| WSE/OSE | 79.9 | 57.8 | 95.8 | 97.7 | 95.5 | 70.5 |
| VRF (ours) | 79.8 | 58.5 | 95.8 | 98.4 | 95.4 | 71.4 |

Conversely, when the optimal OOD value is reached at $\alpha = 0.3$, the performance on ID diminishes by 2.7% from its best. In contrast, our VRF simultaneously attains the ID and OOD performance.

We also analyze the relation between the ratio $\frac{\text{Acc}_{\text{ft}}}{\text{Acc}_{\text{zs}}}$ and $d(\mathbf{x})$ in Figure 3 (a.2&b.2). Consistent with the findings from ImageNet (Figure 1 (b)), we observe that the ratio decreases as $d(\mathbf{x})$ increases, which further supports our design of assigning a higher weight to fine-tuned models if $d(\mathbf{x})$ is smaller.

## 5.3 Further Analysis and Ablation Studies

**VRF for linear-probed models.** A drawback of the proposed method is its doubled inference and storage cost compared to WSE and other single-model robust fine-tuning methods. To address concerns regarding space-time complexity, we apply our VRF method to linear-probed models and present the results in Table 4. We also compare with output-space ensembling, since the model is linear, it is equivalent to weight-space ensembling. We also compare it with output-space ensembling, which, given the linear nature of the model, is equivalent to weight-space ensembling. Consistent with the conclusions drawn from fully fine-tuned models, our VRF method further improves OOD performance while maintaining comparable ID performance to OSE/WSE ensembling.

**Using ZSF set $\mathcal{V}$ or entire training set $\mathcal{D}$?** In Step 1 of our VRF, we define the zero-shot failure set $\mathcal{V}$ and use it to compute distances. We aim to find out whether using the entire training set $\mathcal{D}$ offers comparable performance. In Figure 4, we plot the $\frac{\text{Acc}_{\text{ft}}}{\text{Acc}_{\text{zs}}}$ curves and report both ID and OOD accuracy using the two sets. We observe that the ratio curve using $\mathcal{D}$ does not exhibit a monotonic trend with $d(\mathbf{x})$: it initially increases and then decreases as $d(\mathbf{x})$ increases. Furthermore, the ratio $\frac{\text{Acc}_{\text{ft}}}{\text{Acc}_{\text{zs}}}$ using $\mathcal{D}$ is less informative when $d(\mathbf{x})$ is smaller than 1.2, as the curve relatively remains flat. As the zero-shot models can accurately predict a large proportion of the ID data (recall that

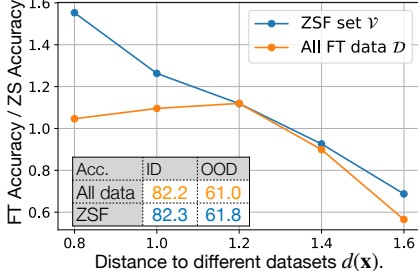

Figure 4: ZSF set $\mathcal{V}$ vs. all data $\mathcal{D}$

the zero-shot accuracy is 68.3%), a smaller distance to entire training set $\mathcal{D}$ does not reliably indicate whether the fine-tuned model can make more accurate predictions. In comparison, our ZSF set contains only the samples where zero-shot models fail but fine-tuned models succeed. When a sample is close to $\mathcal{V}$, it is more likely that the accuracy ratio will be high. Consequently, the performances using $\mathcal{D}$ are clearly outperformed by those using $\mathcal{V}$.

**Comparison with selective prediction using OOD detector.** A simple approach to address the ID-OOD trade-offs is to use an OOD detector for selective prediction. The OOD detector is a binary classifier $G_\lambda(\cdot)$ to decide whether a sample is ID or OOD based on a threshold $\lambda$. For a test sample, predictions are made with the fine-tuned model if classified as ID, and with the zero-shot model otherwise:

$$f_{\text{sp}}(\mathbf{x}) = \begin{cases} f_{\text{ft}}(\mathbf{x}) & \text{if } G_\lambda(\mathbf{x}) = \text{ID} \\ f_{\text{zs}}(\mathbf{x}) & \text{if } G_\lambda(\mathbf{x}) = \text{OOD}, \end{cases} \quad (12)$$

$$G_\lambda(\mathbf{x}) = \begin{cases} \text{ID} & \text{if } S(\mathbf{x}) \geq \lambda \\ \text{OOD} & \text{if } S(\mathbf{x}) < \lambda, \end{cases} \quad (13)$$

Table 5: Selective prediction using OOD detector.

| Method | ID | OOD |
|---|---|---|
| MSP [8] | 81.5 | 57.3 |
| Energy [18] | 81.0 | 57.6 |
| MD [16] | 81.0 | 57.7 |
| kNN [24] | 80.8 | 58.4 |
| RMD [22] | 81.1 | 58.4 |
| VRF (ours) | 82.3 | 61.8 |

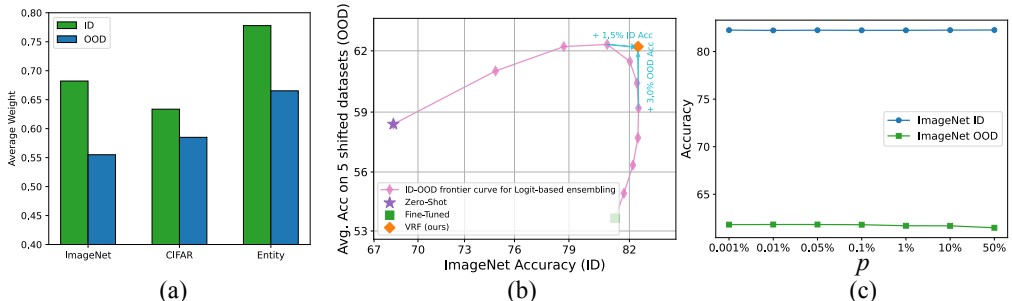

Figure 5: (a) Averaged weight $\mathbb{E}_{\mathbf{x}}[\omega(\mathbf{x})]$ on different datasets. (b) VRF based on logit-space ensembling. (c) Comparison with the effect of different $k$ in the $k$-NN distance.

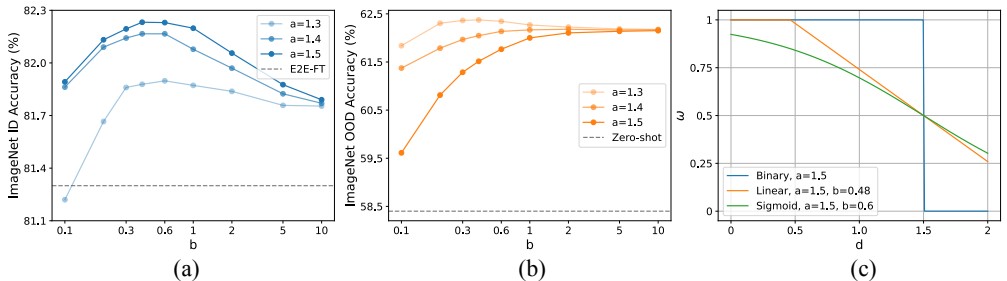

Figure 6: (a) Effect of $a$ and $b$ on ImageNet ID accuracy. (b) Effect of $a$ and $b$ on ImageNet OOD accuracy. (c) Other designs of $\omega(\mathbf{x})$, hyper-parameters are searched on validation set.

where instances with higher scores $S(\mathbf{x})$ are classified as ID and vice versa. $\lambda$ is typically chosen to achieve achieve a $95\%$ true positive rate for ID samples. We report the results of several implementations of $S(\mathbf{x})$ in Table 5 (Details are in Section C.6). We note that selective prediction achieves comparable ID performance to E2E-FT models and similar OOD performance to zero-shot models. However, its accuracy still falls significantly short of our VRF. This is because traditional OOD detectors are designed for scenarios where the OOD data have a completely disjoint label space from the ID data, *i.e.*, $\mathcal{Y}_{\text{OOD}} \cap \mathcal{Y}_{\text{ID}} = \emptyset$. However, in our setup, the zero-shot models show predictive power on ID data, and the fine-tuned models are effective on OOD data. Making binary selections may overlook the complementary knowledge from the other model. Instead, our weight function $\omega(\mathbf{x})$ intelligently selects the contribution of each model based on the distance to the ZSF set. Another reason why our method outperforms selective prediction is the effective use of the ZSF set, as illustrated in Figure 4. Directly using all ID data as traditional OOD detectors (e.g., kNN and MD) leads to a weak correlation between the accuracy ratio and the distance $d(\mathbf{x})$ (or score $S(\mathbf{x})$)

**Examination of the averaged weight for ID and OOD test sets.** Figure 5(a) shows the average weight ($\mathbb{E}_{\mathbf{x}}[\omega(\mathbf{x})]$) of the E2E-FT model in ensembling for both ID and OOD test sets. As expected, higher average weights are observed in the ID test set, as the fine-tuned models excel in such domain.

**Logits-based ensembling.** In this paper, we implement OSE by linearly interpolating the probabilities of the two models. Another common strategy for ensembling, known as Logits-Space Ensembling (LSE), involves interpolating in the logits space: $f(\mathbf{x}; \theta_{\text{lse}}) = \alpha f(\mathbf{x}; \theta_{\text{ft}}) + (1 - \alpha) f(\mathbf{x}; \theta_{\text{zs}})$. We aim to investigate whether our VRF can enhance the robustness of LSE without compromising the ID accuracy. The results depicted in Figure 5(b) confirm that our VRF can indeed generalize to LSE.

**Effect of $k$ in $k$-NN distance.** In Figure 5(c), to compute $k = \text{floor}(p \cdot |\mathcal{V}|)$, we vary $p$ across the range $\{0.0001\%, 0.01\%, 0.05\%, 0.1\%, 10\%, 50\%\}$. We note two observations: (1) Varying $k$ slightly affect the ID performance: the fluctuations are less than $0.1\%$. (2) The OOD accuracy declines as $p$ increases, but the degradation is very slight for relative small values of $p$ (*e.g.*, when $p < 0.01\%$, the decline is smaller than $0.2\%$). In our implementation, we use the $k$-th nearest sample instead of the nearest one to reduce the potential impact of label noise. If the nearest sample is mislabeled, the distance may be unreliable. The $k$-th sample, being in a higher-density region, offers more stable distance estimates with lower variance, as it lies between the $(k-1)$-th and $(k+1)$-th samples. This makes the measure more robust to outliers. Additionally, prior research [24] shows that using the $k$-th nearest distance improves density estimation, which we adopt here.

Table 6: Accuracy of designs of $\omega$ on ImageNet.

| Design of $\omega$ | ID | OOD |
|---|---|---|
| Binary | 81.3 | 58.4 |
| Linear | 82.3 | 61.7 |
| Sigmoid | 82.3 | 61.8 |

Smallest $d(\mathbf{x})$ 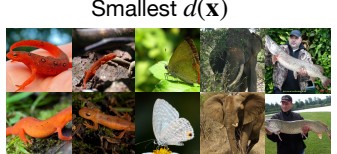  Largest $d(\mathbf{x})$ 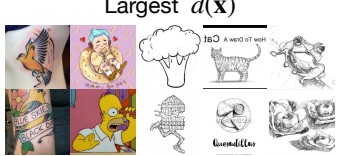

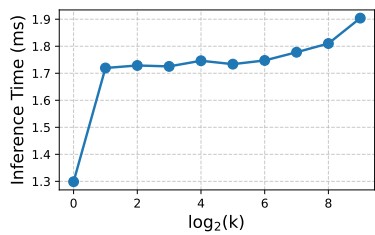

Figure 7: Visualization the samples with the smallest/largest $d(\mathbf{x})$.

**Inference speed of computing $k$-nearest neighbor distance.** Thanks to the Faiss library [11], the $k$-NN search can be efficiently implemented. When evaluated on ImageNet benchmarks using CLIP ViT/B-16 features, the inference speed is approximately 1.8 milliseconds per-image, which does not significantly improve the inference time. In Figure 8, we further present the per-image inference speed of the $k$-nearest neighbor distance computation for various $k$ values. The inference speed is less than 2 ms when $k < 512$.

Figure 8: Inference speed (per-image) using different $k$.

**Effect of $a$ and $b$ in $\omega$.** We demonstrate the effect of $a$ and $b$ in Figure 6 (a&b). We highlight three trends: (1) ID performance peaks at $b \approx 0.6$ across different values of $a$. (2) OOD performance often improves as $b$ increases across different values of $a$. (3) When $b$ is sufficiently large, *e.g.*, $b > 10$ for ID and $b > 2$ for OOD, $a$ has marginal effect on the performance of ID and OOD. In Appendix C.2, we further plot the trade-offs when tuning $a$ and $b$.

**Other designs of $\omega(\mathbf{x})$.** We further explore alternative designs of $\omega(\mathbf{x})$ beyond the sigmoid format in Eq. (6):

- Binary weight: $\omega_{\text{binary}}(\mathbf{x}) = \mathbb{1}[d(\mathbf{x}) < a]$, where $a \in [0, 2]$ and $\mathbb{1}[\cdot]$ is the indicator function.
- Linear weight: $\omega_{\text{linear}}(\mathbf{x}) = \text{clamp}_{[0,1]}(-b \cdot (d(\mathbf{x}) - a))$, where $a \in [0, 2]$, $b > 0$ and $\text{clamp}_{[0,1]}(\cdot)$ rectifies the weight within $[0, 1]$.

We report the results on ImageNet in Table 6 and plot the weight curves with the value of hyperparameters in Figure 6(c). We find that the Linear and the Sigmoid weights show comparable performance and assign similar values of $\omega$ around $d = 1.5$.

**Visualization of samples x according to $d(\mathbf{x})$.** In Figure 7, we randomly sample testing images with the top-100 smallest $d(\mathbf{x})$ values in the range $[0.40, 0.62]$ and the top-100 largest $d(\mathbf{x})$ values in the range $\in [1.59, 1.63]$. Interesting, we observe that: (1) Samples with the smallest $d(\mathbf{x})$ predominantly consist of fine-grained species, *e.g.*, "Triturus vulgaris", "eft" and "lycaenid", where the fine-tuned models possess domain-specific knowledge, which is often lacking in the zero-shot models. (2) Images with the largest $d(\mathbf{x})$ exhibit styles different from those of the fine-tuning samples, including tattoos, cartoons, and sketches, contrasting with the photos typically seen in fine-tuning. Zero-shot models are more skilled in non-real photo styles compared to fine-tuned models.

## 6 Impact, limitations and conclusion

**Impact.** Zero-shot models inherit the weaknesses from pre-training data to downstream tasks, such as noisy and malicious samples. Our VRF might propagate the negative impact.

**Limitations.** Our approach is built on the premise that zero-shot models posses predictive capabilities for downstream tasks. However, if the pre-training knowledge significantly differs from the downstream tasks, our algorithm might fail, which is also an open problem in transfer learning. In addition, the proposed method doubles inference cost compared to WSE and other fine-tuning methods, as it runs both the zero-shot and fine-tuned models. However, this overhead can be mitigated by parallel execution.

**Conclusion.** Inspired by the ID-OOD trade-offs in ensemble-based fine-tuning, we propose VRF to simultaneously optimize the best ID and OOD accuracy. By leveraging the distance to the ZSF set, we assign sample-wise weights to the two models. Despite its simplicity, our VRF demonstrates strong empirical performance, offering a promising technique for solving ID-OOD trade-offs.

## Acknowledgments

This research is supported by the National Research Foundation, Singapore under its AI Singapore Programme (AISG Award No: AISG2-PhD-2021-01-002).

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

# Contents

## A Licenses

All the datasets we considered are publicly available, we list their licences and URLs as follows:

- **CIFAR-10** [13]: MIT License, `https://www.cs.toronto.edu/~kriz/cifar.html`.
- **STL-10** [2]: Non-commercial, `https://cs.stanford.edu/~acoates/stl10/`.
- **Entity-30** [23]: Non-commercial, `https://github.com/MadryLab/BREEDS-Benchmarks`.
- **ImageNet** [3]: Non-commercial, `http://image-net.org`.
- **IN-V2** [21]: MIT License, `https://github.com/modestyachts/ImageNetV2`.
- **IN-R** [7]: MIT License, `https://github.com/hendrycks/imagenet-r`.
- **IN-Sketch** [27]: MIT License, `https://github.com/HaohanWang/ImageNet-Sketch`.

Table 7: Hyper-parameters $a$ and $b$ for different backbones and datasets.

| Backbone | ImageNet | | CIFAR-10 | | Entity-30 | |
|---|---|---|---|---|---|---|
| | $a$ | $b$ | $a$ | $b$ | $a$ | $b$ |
| CLIP ViT-B/32 | 1.5 | 0.6 | 0.3 | 0.3 | 1.1 | 0.6 |
| CLIP ViT-B/16 | 1.5 | 0.5 | 0.3 | 0.3 | 1.1 | 0.6 |

- **IN-A** [9]: MIT License, `https://github.com/hendrycks/natural-adv-examples`.
- **ObjectNet** [1]: Creative Commons Attribution 4.0, `https://objectnet.dev`.

## B   Analysis in the Presence of Correlated Errors

Our assumption of independent residual errors is based on the previous studies [28] (Section 5), where an empirical phenomena is observed: the zero-shot and the fine-tuned models produce diverse predictions. In general (*i.e.*, the fine-tuned models are initialized from the zero-shot models), we cannot assume that the errors in the zero-shot and fine-tuned models are totally uncorrelated. Let $\mathbb{C}[\eta_{zs}(\mathbf{x}), \eta_{ft}(\mathbf{x})]$ be the covariance between $\eta_{zs}(\mathbf{x})$ and $\eta_{ft}(\mathbf{x})$, the variance of $\eta_{vrf}(\mathbf{x})$ can be expressed as:

$$\mathbb{V}[\eta_{vrf}(\mathbf{x})] = g_{zs}(\mathbf{x})^2 \cdot \mathbb{V}[\eta_{zs}(\mathbf{x})] + g_{ft}(\mathbf{x})^2 \cdot \mathbb{V}[\eta_{ft}(\mathbf{x})] + 2 \cdot g_{zs}(\mathbf{x}) \cdot g_{ft}(\mathbf{x}) \cdot \mathbb{C}[\eta_{zs}(\mathbf{x}), \eta_{ft}(\mathbf{x})]. \quad (14)$$

Maintaining that $g_{zs}(\mathbf{x}) + g_{ft}(\mathbf{x}) = 1$, the optimal weight $g_{ft}^*(\mathbf{x})$ to minimize Eq. (14) becomes:

$$g_{ft}^*(\mathbf{x}) = (1 + \frac{\mathbb{V}[\eta_{ft}(\mathbf{x})] - \mathbb{C}[\eta_{zs}(\mathbf{x}), \eta_{ft}(\mathbf{x})]}{\mathbb{V}[\eta_{zs}(\mathbf{x})] - \mathbb{C}[\eta_{zs}(\mathbf{x}), \eta_{ft}(\mathbf{x})]})^{-1} \quad (15)$$

Recall that we are interested in using the distance to ZSF set, *i.e.*, $d(\mathbf{x})$, to surrogate $g_{ft}^*(\mathbf{x})$. To understand the relationship between $d(\mathbf{x})$ and $g_{ft}^*(\mathbf{x})$, we first group test samples in ImageNet and its five distribution shifted datasets into bins based on the value of $d(\mathbf{x})$. We then compute the averaged $g_{ft}^*(\mathbf{x})$ for each bin and plot the relationship in Figure 9. In specific, we use temperature scaling [14] to calibrate the zero-shot and fine-tuned models over the validation set. Afterwards, we define the true distribution $\mathbb{P}(y|\mathbf{x})$ as a one-hot vector, where the value of 1 corresponds to the true label for a given input

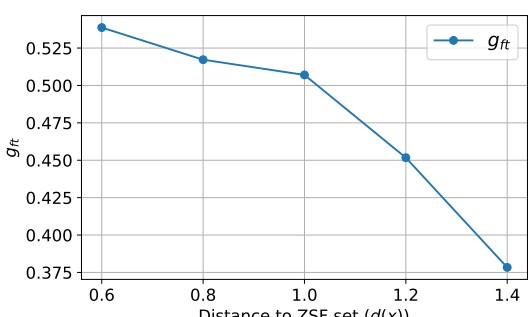

Figure 9: Relationship between $g_{ft}(\mathbf{x})$ and $d(\mathbf{x})$.

$\mathbf{x}$. We then calculate $\eta_{ft}(\mathbf{x}) = \hat{\mathbb{P}}(y|\mathbf{x}; \theta_{ft}) - \mathbb{P}(y|\mathbf{x})$ and $\eta_{zs}(\mathbf{x}) = \hat{\mathbb{P}}(y|\mathbf{x}; \theta_{zs}) - \mathbb{P}(y|\mathbf{x})$. Finally, we compute the average $g_{ft}^*(\mathbf{x})$ for each bin as shown in Figure 9. Interestingly, we observe the similar trend in Figure 1 (b): the weight for fine-tuned models decreases as $d(\mathbf{x})$ increases. This phenomena indicates that our weighting function $\omega(\mathbf{x})$ derived under the assumption of independent errors is also valid in the presence of correlated errors.

## C   Additional Experimental Details and Results

### C.1   Additional Experimental Details

For CLIP ViT-32 based E2E-FT and LP-FT models, we use a batch size of 512. For CLIP ViT-16 based E2E-FT, we directly download the fine-tuned models from Wortsman et al. [28][1]. The batch size for training CLIP ViT-16 based LP-FT models is set to 384, which is the largest batch size that fits into 2 A6000 GPUs. When performing linear probing, we use a batch of 512 and the initial learning rate of 0.1 for all experiments. The mixing coefficient $\alpha$ for OSE and WSE are searched over $[0, 0.1, 0.2, ..., 0.9, 1.0]$. The values of $a$ and $b$ for our VRF are reported in Table 7.

---

[1]`https://drive.google.com/drive/folders/1f56kjpRKPiNSaUxNDtETEDRkbDkZnpCQ`

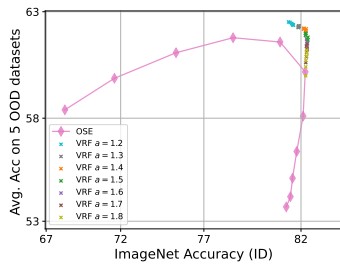 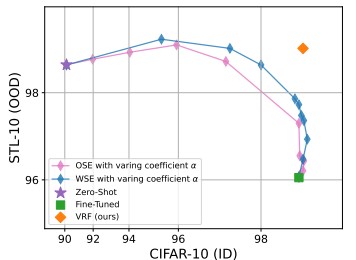 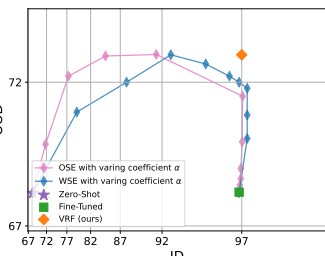

Figure 10: ID-OOD scatters for VRF on ImageNet and its variants.

Figure 11: ID-OOD frontier curves for OSE and WSE for CIFAR-10 → STL10.

Figure 12: ID-OOD frontier curves for OSE and WSE for Entity-30.

Table 8: Accuracy of E2E-FT based OSE on ImageNet and derived distribution shifts for various values of the mixing coefficient $\alpha$. Results shown for CLIP ViT-B/16.

| $\alpha$ | IN | Distribution shifts | | | | | Avg shifts |
|---|---|---|---|---|---|---|---|
| | | IN-V2 | IN-Sketch | IN-A | IN-R | ObjectNet | |
| 0.0 | 68.3 | 61.9 | 48.3 | 50.1 | 77.6 | 54.2 | 58.4 |
| 0.1 | 71.6 | 64.8 | 49.9 | 50.7 | 78.5 | 55.5 | 59.9 |
| 0.2 | 75.4 | 67.5 | 51.6 | 51.2 | 79.2 | 56.3 | 61.1 |
| 0.3 | 78.6 | 69.7 | 52.3 | 50.9 | 79.3 | 56.8 | 61.8 |
| 0.4 | 81.0 | 71.3 | 52.0 | 49.6 | 78.6 | 56.6 | 61.6 |
| 0.5 | 82.2 | 72.0 | 50.6 | 46.8 | 76.7 | 54.9 | 60.2 |
| 0.6 | 82.1 | 71.6 | 48.7 | 42.9 | 73.8 | 53.3 | 58.1 |
| 0.7 | 81.8 | 71.2 | 47.3 | 40.5 | 71.1 | 52.2 | 56.4 |
| 0.8 | 81.6 | 70.9 | 46.3 | 38.5 | 68.3 | 51.4 | 55.1 |
| 0.9 | 81.5 | 70.7 | 45.6 | 37.4 | 66.6 | 50.8 | 54.2 |
| 1.0 | 81.3 | 70.6 | 45.1 | 36.6 | 65.6 | 50.5 | 53.7 |

## C.2 Plotting ID-OOD Trade-Offs of VRF

In Figure 10, we present an ID-OOD scatter plot over a wide range of $a$ and $b$ values on ImageNet and its five variants using CLIP ViT-B/16. Specifically, $a$ varies from 1.2 to 1.8, while $b$ ranges from 0.5 to 1.0. Our method consistently achieves better ID-OOD trade-offs, as indicated by its points lying outside the OSE curve (represented by the magenta curve) across different configurations.

## C.3 Additional Results for OSE, WSE and VRF with Varying Hyper-Parameters

Results for all mixing coefficient $\alpha$ for OSE and WSE are available in Table 8 and Table 9, respectively. Results for values of $a$ and $b$ are available in Table 10. In addition, we plot the ID-OOD trade-off curves for OSE and WSE on the CIFAR-10 and Entity-30 datasets in Figures 11 and 12, respectively.

## C.4 Optimal Performance Searched on Test Sets

We have conducted additional experiments where we optimized the hyperparameters for each test set of the ImageNet benckmarks. The results are summarized in Table 11.

## C.5 Combining VRF with Other Robust Fine-Tuning Methods

Our VRF framework is orthogonal and complementary to existing fine-tuned models. To demonstrate this, we integrated FLYP [5] into our VRF framework. The results in Table 12 show that VRF improves OSE's performance under distribution shift by 1.1% without compromising in-distribution (ID) performance.

## C.6 Additional Results for Selective Prediction Using OOD Detectors

We provide the breakdown performance for selective prediction using OOD detectors in Table 13.

Table 9: Accuracy of E2E-FT based WSE on ImageNet and derived distribution shifts for various values of the mixing coefficient $\alpha$. Results shown for CLIP ViT-B/16.

| $\alpha$ | IN | Distribution shifts | | | | | Avg shifts |
|---|---|---|---|---|---|---|---|
| | | IN-V2 | IN-Sketch | IN-A | IN-R | ObjectNet | |
| 0.0 | 68.3 | 61.9 | 48.3 | 50.1 | 77.6 | 54.2 | 58.4 |
| 0.1 | 72.9 | 65.7 | 50.8 | 52.5 | 79.4 | 55.7 | 60.8 |
| 0.2 | 76.4 | 68.7 | 52.5 | 54.2 | 80.1 | 57.1 | 62.5 |
| 0.3 | 78.9 | 70.6 | 53.6 | 54.6 | 80.1 | 57.5 | 63.3 |
| 0.4 | 80.5 | 72.1 | 54.1 | 53.8 | 79.6 | 57.7 | 63.5 |
| 0.5 | 81.7 | 72.8 | 53.9 | 52.2 | 78.7 | 57.3 | 63.0 |
| 0.6 | 82.4 | 72.9 | 53.4 | 50.0 | 77.2 | 56.2 | 61.9 |
| 0.7 | 82.5 | 73.2 | 52.4 | 47.4 | 75.2 | 55.0 | 60.6 |
| 0.8 | 82.5 | 72.8 | 51.0 | 44.6 | 72.7 | 53.5 | 58.9 |
| 0.9 | 82.1 | 72.0 | 48.9 | 40.9 | 69.5 | 51.7 | 56.6 |
| 1.0 | 81.3 | 70.6 | 45.1 | 36.6 | 65.6 | 50.5 | 53.7 |

Table 10: Accuracy of E2E-FT based VRF on ImageNet and derived distribution shifts for various values of $a$ and $b$. Results shown for CLIP ViT-B/16.

| $a$ | $b$ | IN | Distribution shifts | | | | | Avg shifts |
|---|---|---|---|---|---|---|---|---|
| | | | IN-V2 | IN-Sketch | IN-A | IN-R | ObjectNet | |
| 1.4 | 0.5 | 82.2 | 72.2 | 52.7 | 49.6 | 79.4 | 56.7 | 62.1 |
| 1.4 | 0.6 | 82.2 | 72.2 | 52.7 | 49.6 | 79.4 | 56.7 | 62.1 |
| 1.4 | 0.7 | 82.2 | 72.2 | 52.7 | 49.7 | 79.4 | 56.8 | 62.2 |
| 1.4 | 0.8 | 82.1 | 72.2 | 52.7 | 49.7 | 79.4 | 56.8 | 62.2 |
| 1.4 | 0.9 | 82.1 | 72.1 | 52.7 | 49.7 | 79.4 | 56.8 | 62.2 |
| 1.4 | 1.0 | 82.1 | 72.1 | 52.7 | 49.7 | 79.4 | 56.8 | 62.2 |
| 1.5 | 0.5 | 82.3 | 72.1 | 52.3 | 48.7 | 79.0 | 56.2 | 61.7 |
| 1.5 | 0.6 | 82.3 | 72.1 | 52.4 | 48.9 | 79.1 | 56.4 | 61.8 |
| 1.5 | 0.7 | 82.2 | 72.2 | 52.4 | 49.1 | 79.2 | 56.5 | 61.9 |
| 1.5 | 0.8 | 82.2 | 72.2 | 52.5 | 49.2 | 79.2 | 56.6 | 61.9 |
| 1.5 | 0.9 | 82.2 | 72.2 | 52.5 | 49.3 | 79.2 | 56.6 | 62.0 |
| 1.5 | 1.0 | 82.2 | 72.2 | 52.6 | 49.4 | 79.2 | 56.6 | 62.0 |
| 1.6 | 0.5 | 82.3 | 71.9 | 51.9 | 48.0 | 78.5 | 55.7 | 61.2 |
| 1.6 | 0.6 | 82.3 | 72.1 | 52.1 | 48.3 | 78.6 | 55.9 | 61.4 |
| 1.6 | 0.7 | 82.3 | 72.1 | 52.2 | 48.5 | 78.8 | 56.0 | 61.5 |
| 1.6 | 0.8 | 82.3 | 72.2 | 52.3 | 48.6 | 78.9 | 56.1 | 61.6 |
| 1.6 | 0.9 | 82.3 | 72.2 | 52.3 | 48.7 | 79.0 | 56.2 | 61.7 |
| 1.6 | 1.0 | 82.3 | 72.1 | 52.4 | 48.8 | 79.0 | 56.3 | 61.7 |

## C.7 Curves of $\frac{\text{Acc}_{\text{ft}}}{\text{Acc}_{\text{zs}}}$ for ImageNet and its Five Distribution Shifted Datasets

In Figure 13, we examine the relationship between $\frac{\text{Acc}_{\text{ft}}}{\text{Acc}_{\text{zs}}}$ and $d(\mathbf{x})$ for ImageNet and its five derived distribution shifted datasets. Based on the value of $d(\mathbf{x})$, test samples are grouped into bins, and we compute the ratio of fine-tuned accuracy to zero-shot accuracy for each bin. For example, to compute the value of $\frac{\text{Acc}_{\text{ft}}}{\text{Acc}_{\text{zs}}}$ at $d(\mathbf{x}) = 0.8$, we first identify the samples with $d(\mathbf{x}) \in [0.7, 0.9]$, then compute the averaged accuracy for these samples using zero-shot models and fine-tuned models, and finally

Table 11: Optimal Results search on test set on ImageNet and its five variants for CLIP ViT-B/16.

| Method | IN | Distribution shifts | | | | | Avg shifts |
|---|---|---|---|---|---|---|---|
| | | IN-V2 | IN-Sketch | IN-A | IN-R | ObjectNet | |
| E2E-FT | 81.3 | 70.6 | 45.1 | 36.6 | 65.6 | 50.5 | 53.7 |
| + VRF (ours) | 82.3 | 72.1 | 52.9 | 48.4 | 78.7 | 56.4 | 61.8 |
| + VRF (oracle) | 82.3 | 72.2 | 53.0 | 51.4 | 79.7 | 57.9 | 62.9 |

Table 12: Applying VRF to other robust fine-tuning methods.

| Method | IN | Distribution shifts | | | | | Avg shifts |
| | | IN-V2 | IN-Sketch | IN-A | IN-R | ObjectNet | |
| --- | --- | --- | --- | --- | --- | --- | --- |
| FLYP [5] | 82.6 | 73.0 | 71.4 | 48.1 | 49.6 | 58.7 | 60.2 |
| + WSE | 82.9 | 73.5 | 76.0 | 53.0 | 52.3 | 60.8 | 63.1 |
| + OSE | 82.8 | 73.6 | 77.0 | 52.5 | 51.9 | 59.9 | 62.8 |
| + VRF | 82.8 | 73.6 | 78.6 | 52.9 | 53.0 | 61.2 | 64.0 |

Table 13: Breakdown performance for selective prediction using OOD detector.

| OOD Detector | IN | Distribution shifts | | | | | Avg shifts |
| | | IN-V2 | IN-Sketch | IN-A | IN-R | ObjectNet | |
| --- | --- | --- | --- | --- | --- | --- | --- |
| MSP [8] | 81.5 | 71.1 | 48.6 | 42.1 | 71.6 | 52.9 | 57.3 |
| Energy [18] | 81.0 | 70.5 | 48.1 | 42.3 | 73.9 | 53.0 | 57.6 |
| MD [16] | 81.0 | 70.4 | 49.7 | 41.7 | 74.1 | 52.6 | 57.7 |
| kNN [24] | 80.8 | 70.4 | 49.5 | 43.6 | 74.8 | 53.6 | 58.4 |
| RMD [22] | 81.1 | 70.6 | 49.6 | 44.4 | 74.4 | 53.1 | 58.4 |

compute the ratio. Note that the averaged ratio $\frac{\text{Acc}_{ft}}{\text{Acc}_{zs}}$ on ImageNet-{A,R} and ObjectNet is undefined for $d(\mathbf{x}) = 0.6$. This is because in these datasets, the zero-shot accuracy around $d(\mathbf{x}) = 0.6$ is 0. We observe that the trend of the ratio $\frac{\text{Acc}_{ft}}{\text{Acc}_{zs}}$ decreasing as $d(\mathbf{x})$ increasing is stable for all ImageNet related datasets.

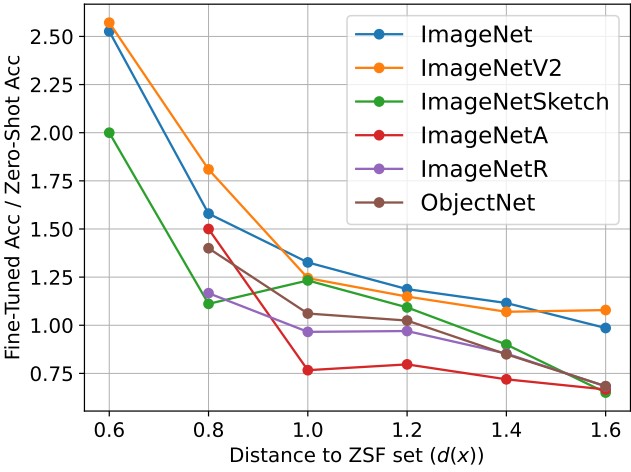

Figure 13: Relationship between $\frac{\text{Acc}_{ft}}{\text{Acc}_{zs}}$ and $d(\mathbf{x})$ on ImageNet benchmarks.

