# OpenReview forum: "Robust Fine-tuning of Zero-shot Models via Variance Reduction"
_NeurIPS.cc/2024/Conference — NeurIPS 2024 poster_

### Official Review · Reviewer_ydTd · 2024-07-09

**Soundness:** 4
**Presentation:** 4
**Contribution:** 3
**Rating:** 6
**Confidence:** 4

**Summary:**

This paper addresses the ID-OOD trade-off of the ensemble-based method under the pretrain-finetune regime. The authors propose a simple yet effective adaptive ensemble method that determines the ensemble coefficient based on the feature distance between a test sample and the zero-shot failure set. The proposed method has a theoretical foundation, and it consistently improves the OOD generalization performance compared to the vanilla ensemble baseline in diverse situations.

**Strengths:**

* The paper is well-written overall and easy to follow.
* The research problem this work focuses on is important in the era of foundation models, and the proposed method to address this is very straightforward and reasonable.
* The proposed method shows consistent performance improvement in numerous distribution shift setups, and the author provides an intuitive theoretical justification for the proposed method.
* Besides the empirical success in terms of improved ID-OOD performance trade-off, the authors present extensive qualitative and quantitative analysis results that give numerous insights into this field.

**Weaknesses:**

* The method requires 1) accessibility to the entire training dataset and 2) distance computation between each test sample and the entire train sample during inference time. These limit its application to resource-constrained (in terms of accessibility, memory storage, and runtime) situations.
* Limited implication of the theoretical result
  * While the current theoretical analysis provides a justification for the optimal strategy for determining the ensemble coefficients, which is realized by the authors, it does not say anything about the relative generalization error between the proposed method and other non-ensemble-based fine-tuning methods.
* Lack of possible comparison with more advanced fine-tuning [4,5,6,7] or ensemble [1,2,3] methods.
  * While the proposed approach has its unique implication compared to the vanilla ensemble, missing comparison with advanced baselines raises concern about the practical utility of the proposal.

---
> Reference
* [1] SIMPLE: Specialized Model-Sample Matching for Domain Generalization, Li et al. 2023
* [2] Generalized Logit Adjustment: Calibrating Fine-tuned Models by Removing Label Bias in Foundation Models, Zhu et al. 2023
* [3] Pack of LLMs: Model Fusion at Test-Time via Perplexity Optimization, Mavromatis et al. 2024
* [4] Fine-Tuning can Distort Pretrained Features and Underperform Out-of-Distribution, Kumar et al. 2022
* [5] Finetune like you pretrain: Improved finetuning of zero-shot vision models, Goyal et al. 2023
* [6] Trainable Projected Gradient Method for Robust Fine-tuning, Tian et al. 2023
* [7] Towards Calibrated Robust Fine-Tuning of Vision-Language Models, Oh et al. 2024

**Questions:**

* Could the proposed method also be applied to the weight-space ensemble rather than the output-space ensemble?
* How sensitive is the proposed method against varying numbers of training samples that determine the quality of the zero-shot failure set?

**Limitations:**

The authors adequately stated the limitation in the paper.

---

> ### Author Rebuttal · Authors · 2024-08-06
>
> **Q1**:  Weaknesses on computation costs.
>
> **A1**: For efficiently performing k-NN search, we use Faiss library [11], which can perform billion-scale similarity search with GPUs. For our ImageNet experiments, the inference speed of kNN search for a single image is averaged 3.2 ms. For our ImageNet experiments, the storage size for the ZSF set features is 289 MB. While this does require additional storage compared to some traditional methods, we believe it is a manageable overhead given the benefits in robustness and performance that our VRF framework provides.
>
> **Q2**: Limited implication of the theoretical result.
>
> **A2**: In non-ensemble-based fine-tuning methods, the model can be seen as using fixed coefficients, essentially $g_{ft}(x) = 1$. Since this is not the optimal strategy for balancing ID and OOD performance, the generalization error for non-ensemble-based fine-tuning is expected to be relatively larger compared to our VRF method, which optimizes these coefficients.
>
> **Q3**: Lack of possible comparison with more advanced fine-tuning.
>
> **A3**: Since our VRF framework is orthogonal to the fine-tuned models, we use the [4][5] as the fine-tuned models and report results below.
> |      | IN    | V2    | R     | A     | S     | ObjectNet | OOD  |
> |------|-------|-------|-------|-------|-------|-----------|------|
> | [4]  | 81.5  | 70.7  | 46.7  | 41.4  | 66.4  | 52.4      | 55.5 |
> | +WSE | 82.4  | 73.0  | 51.5  | 50.6  | 74.2  | 56.6      | 61.2 |
> | +OSE | 82.1  | 72.3  | 50.9  | 50.9  | 74.9  | 55.7      | 60.9 |
> | +VRF | 82.1  | 72.3  | 52.9  | 51.2  | 78.8  | 57.2      | 62.4 |
>
> |      | IN    | V2    | R     | A     | S     | ObjectNet | OOD  |
> |------|-------|-------|-------|-------|-------|-----------|------|
> | [5]  | 82.6  | 73.0  | 71.4  | 48.1  | 49.6  | 58.7      | 60.2 |
> | +WSE | 82.9  | 73.5  | 76.0  | 53.0  | 52.3  | 60.8      | 63.1 |
> | +OSE | 82.8  | 73.6  | 77.0  | 52.5  | 51.9  | 59.9      | 62.8 |
> | +VRF | 82.8  | 73.6  | 78.6  | 52.9  | 53.0  | 61.2      | 64.0 |
>
> **Q4**: Could the proposed method also be applied to the weight-space ensemble?
>
> **A4**: Yes, our method can be applied to weight-space ensemble models. To do this efficiently, we first generate a set of weight-ensembled models with varying coefficients $\alpha=[0.1,0.2,..,0.9]$. For each sample, we calculate the weight function and select the cached WSE model with the closest $\alpha$. The results for CIFAR10 -> STL10 validate that our VRF approach is effective for weight-space ensemble methods as well.
>
> |             | ID   | OOD  |
> |-------------|------|------|
> | VRF for OSE | 98.6 | 98.4 |
> | VRF for WSE | 98.5 | 98.7 |
>
> **Q5**: How sensitive is the proposed method against varying numbers of training samples that determine the quality of the zero-shot failure set?
>
> **A5**: Our proposed method is robust to variations in the number of training samples in the zero-shot failure set. To validate this, we randomly downsampled the zero-shot failure set by 10%, 20%, and 50%. The results showed only minor fluctuations of ID and OOD performances. This demonstrates that our method maintains its effectiveness even with a reduced zero-shot failure set.
>
> |     | 10%  | 20%  | 50%  | 100% |
> |-----|------|------|------|------|
> | ID  | 82.2 | 82.4 | 82.5 | 82.3 |
> | OOD | 61.8 | 62.0 | 61.7 | 61.8 |

---

> > ### Comment · Reviewer_ydTd · 2024-08-08
> >
> > Thanks for the kind answers!
> >
> > Overall concerns are addressed by the author, while I still wonder about the relative superiority compared with another input-dependent advanced ensemble method (but whether this is addressed or not, I will keep my rating).
> > Anyway, the proposed method is quite novel and effective and has a good theoretical background.
> > I am looking forward to seeing this paper as an official publication soon!
> >
> > reviewer ydTd

---

> > > ### Author Response · Authors · 2024-08-08
> > > **Response to Reviewer ydTd**
> > >
> > > Thank you for the positive feedback and valuable comments. We appreciate your interest in comparing our method with other ensemble methods and we will consider it for future work.  We are delighted that you find our method novel and effective, and we look forward to the possibility of sharing this work as an official publication soon.

---

### Official Review · Reviewer_52xu · 2024-07-09

**Soundness:** 2
**Presentation:** 2
**Contribution:** 3
**Rating:** 5
**Confidence:** 3

**Summary:**

This study examines the robust fine-tuning of CLIP models. The proposed method, Variance Reduction Fine-tuning (VRF), employs a sample-wise ensembling technique to enhance both ID and OOD accuracy, reducing trade-offs between them. Experimental results on ImageNet and associated distribution shifts empirically validate the effectiveness of this approach.

**Strengths:**

- The proposed approach is straightforward, combining outputs from zero-shot and fine-tuned models through output-space ensembling. Unlike conventional methods that use uniform or predefined weighting coefficients, the notable aspect here is the use of different weighting coefficients for each individual instance.

- It is clear that adjusting these weighting coefficients according to whether zero-shot or fine-tuned models are suitable for specific data can improve the final performance. Consequently, the primary challenge lies in devising an effective method to determine such instance-specific weighting coefficients. The authors propose a method that makes use of the Zero-Shot Failure (ZSF) set, consisting of training examples where the zero-shot model fails to predict accurately but the fine-tuned model succeeds. Experimental validation confirms the efficacy of the proposed approach.

- I appreciate Section 5.3 for further presenting an empirical analysis of the proposed method. Essentially, the approach is a variation of OSE, determining how much the zero-shot model and the fine-tuned model are used on a per-instance basis, and VRF is not the only strategy to achieve this. It is gratifying to see that VRF outperforms other alternatives and that the analysis examines how different VRF designs produce varying outcomes.

**Weaknesses:**

- Although there has been considerable research on robust fine-tuning for CLIP (as noted by the authors referencing various studies), the baselines analyzed in Tables 1 and 2 are quite limited. A key advantage of the VRF method is its applicability to any (zero-shot, fine-tuned) pair, both retrospectively and prospectively. Therefore, applying VRF to other robust fine-tuning methods beyond FT and LP-FT would further demonstrate the effectiveness of the proposed VRF approach.

- The proposed method has a drawback of needing more than double the inference cost compared to WSE and other robust fine-tuning methods that yield a single fine-tuned model, since it requires performing a forward pass through the entire network twice (once for zero-shot and once for fine-tuned models) and calculating the ZSF-based weighting coefficient. However, an analysis of such inference costs has not yet been performed. It is important to note that Wortsman et al. (2022) favored WSE over OSE for combining zero-shot and fine-tuned models, highlighting this choice not only for its improved performance but also for its lack of additional costs.

- In the context of robust fine-tuning, we can only observe performance on in-distribution data. Therefore, the parameters
$a$ and $b$ in the proposed Eq. (6) for sample-wise mixing coefficients should be determined based on the in-distribution validation set, as mentioned by the authors in lines 141 and 198-200. From Appendix C.1, it is understood that $\\alpha \\in \\{1.4, 1.5, 1.6\\}$ and $b \\in \\{0.5, 0.6, ..., 1.0\\}$ were considered. However, it seems that the range of values for $a$ and $b$ is too narrow. Based on the in-distribution results in Table 8, larger values for $a$ should be explored, but the authors did not do so. What happens if the value of $a$ is increased further? Would it still outperform OSE and WSE baselines?

**Questions:**

1. In combining zero-shot and fine-tuned models, WSE offers the advantage of requiring only half the inference cost compared to VRF by maintaining inference costs equivalent to that of a single model through weight averaging. When combining zero-shot and linear-probed models, since the image encoder part, which carries most of the inference cost, only needs to be processed once, it appears that VRF's concerns regarding cost could be mitigated. In this context, could the proposed VRF method also extend to "Linear classifier" models, alongside "E2E-FT" and "LP-FT"? It is worth noting that Wortsman et al. (2021) explores ensembling not only between zero-shot models and end-to-end fine-tuned models ("E2E-FT" in this work) but also between zero-shot models and linear classifier fine-tuned models (referred to as "Linear classifier" here).

2. In Figures 3, does "Ensembling with varying coefficient $\alpha$" refer to OSE? It would be helpful to also provide the curve for WSE.

3. Table 7 shows the results of exploring $\alpha$ values for OSE. It would be beneficial to provide the same results for WSE as well.

4. The proposed VRF approach also produces multiple points on trade-off plots (e.g., Figure 3(a.1)) depending on the values of $a$ and $b$ (especially $a$). It would be advantageous to visualize a scatter plot across a broad range of $a$ and $b$ values, allowing us to observe the trade-off curve similar to OSE and WSE, which interpolate between zero-shot and fine-tuned models.

**Limitations:**

The authors addressed the limitations and potential negative societal impact in Section 6.

---

> ### Author Rebuttal · Authors · 2024-08-07
>
> **Q1**: Applying VRF to other robust fine-tuning methods.
>
> **A1**: To further demonstrate the versatility and effectiveness of our VRF method, we applied it to another robust fine-tuning method, FTYP, and conducted experiments on ImageNet and its variants. As expected, our VRF framework further improves OOD performance without sacrificing ID accuracy, reinforcing its applicability across different fine-tuning strategies. We will include these additional results in the revised version.
>
> |      | IN   | V2   | R    | A    | S    | ObjectNet | OOD  |
> |------|------|------|------|------|------|-----------|------|
> | FLYP | 82.6 | 73.0 | 71.4 | 48.1 | 49.6 | 58.7      | 60.2 |
> | +WSE | 82.9 | 73.5 | 76.0 | 53.0 | 52.3 | 60.8      | 63.1 |
> | +OSE | 82.8 | 73.6 | 77.0 | 52.5 | 51.9 | 59.9      | 62.8 |
> | +VRF | 82.8 | 73.6 | 78.6 | 52.9 | 53.0 | 61.2      | 64.0 |
>
> **Q2**: What happens if the value of  is increased further? Would it still outperform OSE and WSE baselines?
>
> **A2**: For the hyperparameter $a$, we chose a range less than 1.6 based on the nature of the ZSF set of ImageNet, where nearly 99% of validation sample distances were less than 1.6. Exploring larger values of $a$ (i.e., $a>1,6$) is not necessary because it results in almost all predictions being assigned to the fine-tuned models, reducing the contribution of the zero-shot models. Since the features are L2-normalized, the distance d(x) is bounded between [0, 2], we further tested values $a = 1.7, 1.8, 1.9 $ and observed that OOD performance decreased as expected, with the fine-tuned model dominating the ensemble contribution.
>
> For the hyperparameter $b$, as shown in Figure 6(a), we noticed that increasing $b$ beyond 1 leads to a decrease in ID performance. Therefore, we limited our exploration to $b \leq 1 $ to maintain a balance between ID and OOD performance.
> | a   | b   | ID   | OOD  |
> |-----|-----|------|------|
> | 1.5 | 0.5 | 82.3 | 61.8 |
> | 1.7 | 0.5 | 82.2 | 60.6 |
> | 1.7 | 0.7 | 82.2 | 61.1 |
> | 1.7 | 0.9 | 82.3 | 61.3 |
> | 1.7 | 1.0 | 82.2 | 61.3 |
> | 1.8 | 0.5 | 82.2 | 60.0 |
> | 1.8 | 0.7 | 82.3 | 60.7 |
> | 1.8 | 0.9 | 82.2 | 61.1 |
> | 1.8 | 1.0 | 82.2 | 61.2 |
> | 1.9 | 0.5 | 82.1 | 59.4 |
> | 1.9 | 0.7 | 82.2 | 60.3 |
> | 1.9 | 0.9 | 82.2 | 60.8 |
> | 1.9 | 1.0 | 82.2 | 60.9 |
>
> **Q3**: Could the proposed VRF method also extend to "Linear classifier" models?
>
> **A3**: Yes, using linear classifiers can indeed reduce the inference costs, as the image encoder only needs to be processed once, effectively halving the costs compared to VRF's standard approach. To explore this, we trained a linear classifier based on CLIP-ViT/16 models and compared the performance with WSE and our VRF. The results confirmed that our VRF framework effectively addresses the ID-OOD trade-offs, achieving higher OOD performance without compromising ID accuracy. This demonstrates that VRF can extend to linear classifier models, providing a versatile and effective solution for robust fine-tuning.
>
> |                  | ID   | OOD  |
> |------------------|------|------|
> | Linear Classifer | 79.3 | 55.2 |
> | +WSE/OSE         | 79.9 | 57.8 |
> | +VRF             | 80.0 | 61.0 |
>
> **Q4**: In Figures 3, does "Ensembling with varying coefficient " refer to OSE? It would be helpful to also provide the curve for WSE.
>
> **A4**:  Yes, the "Ensembling with varying coefficient" refers to OSE. We have also evaluated the ID and OOD performance with varying $\alpha$ for WSE. The curves are plotted in the attached PDF.
>
> **Q5**:  Table 7 shows the results of exploring  values for OSE. It would be beneficial to provide the same results for WSE as well.
>
> **A5**: Thanks for your suggestions. We provide the WSE results with varying $\alpha$ values as below and we will add them in the revised version.
>
> | $\alpha$ | ID   | OOD  |
> |----------|------|------|
> | 0.0      | 68.3 | 58.1 |
> | 0.1      | 72.9 | 60.8 |
> | 0.2      | 76.4 | 62.5 |
> | 0.3      | 78.9 | 63.3 |
> | 0.4      | 80.5 | 63.4 |
> | 0.5      | 81.7 | 63.0 |
> | 0.6      | 82.4 | 61.9 |
> | 0.7      | 82.5 | 60.6 |
> | 0.8      | 82.5 | 58.9 |
> | 0.9      | 82.1 | 56.6 |
> | 1.0      | 81.3 | 53.8 |
>
> **Q6**: It would be advantageous to visualize a scatter plot across a broad range of  a and b values.
>
> **A6**: We appreciate the suggestions. We have visualized the trade-off curves in the attached PDF.

---

> > ### Comment · Reviewer_52xu · 2024-08-09
> >
> > Thank you to the authors for addressing most of my concerns. Would it be possible to revise Figure 12 to include WSE, with Zero-Shot and Fine-Tuned as the endpoints, consistent with the other figures?

---

> > > ### Author Response · Authors · 2024-08-09
> > >
> > > We appreciate the reviewer's advice. Including the WSE frontier curve in Figure 12 is consistent with Figures 10 and 11. As the attached PDF is not editable, we are pleased to add the curve in the revised version.
> > >
> > > Sincerely,
> > > The authors of Submission 3218

---

> > > > ### Comment · Reviewer_52xu · 2024-08-10
> > > >
> > > > The authors have addressed most of my concerns in their rebuttal, including: 1) combining VRF with other robust fine-tuning methods, 2) using a linear probed model instead of a fine-tuned one, and 3) presenting ID-OOD trade-off plots.
> > > >
> > > > Regarding the third point, I recommend that VRF be shown as a curve connecting the zero-shot and fine-tuned points on the trade-off plots, rather than as a single point. Since VRF, like OSE and WSE, is a post-hoc method that combines zero-shot and fine-tuned models, the authors should simply demonstrate that it appears in the top right position on the ID-OOD trade-off plots compared to OSE and WSE. Personally, I sense that the authors may be reluctant to show VRF underperforming the baseline when an inappropriate value of $a$ is chosen, but this is a natural outcome and does not need to be avoided (as the two extremes of $a$ would correspond to either the zero-shot or fine-tuned model).
> > > >
> > > > While I have not received a response to the second weakness I raised, I reviewed the authors' comments to other reviewers. It seems that other reviewers have also expressed concerns about the additional computational burdens introduced by VRF. The current explanation is somewhat insufficient (which I understand might be due to the limited time for the rebuttal), and I believe the authors should provide a thorough analysis of this issue in the revised version. Therefore, if the revised version includes more detailed information on the ID-OOD trade-off plots and computational cost analysis (in addition to addressing the other issues mentioned in the rebuttal), I am willing to raise my score to 5.

---

> > > > > ### Author Response · Authors · 2024-08-10
> > > > >
> > > > > Dear Reviewer 52xu,
> > > > >
> > > > > Thank you for your thoughtful comments. We agree with your concerns about the impact of inappropriate choices for $a$ on performance. As you mentioned, setting $a$ too high biases the final predictions toward the fine-tuned models, while setting it too low favors the zero-shot models. In the revised version, we will address this issue directly and emphasize the need for careful selection of $a$.
> > > > >
> > > > > Regarding the use of the linear probed model, we have provided detailed results in **A3**. With a fixed backbone, OSE and WSE achieved 79.9% ID accuracy and 57.8% OOD accuracy. Our VRF further improved these results, reaching 80.0% ID accuracy and 61.0% OOD accuracy.
> > > > >
> > > > > We appreciate your willingness to raise the score to 5. We will seriously take the reviewer’s advice to revise the paper and  promise that the revised paper will thoroughly address all concerns.
> > > > >
> > > > > Sincerely,
> > > > > Authors of Submission 3218

---

> ### Comment · Reviewer_52xu · 2024-08-10
>
> I have updated my rating to 5 based on the authors' assurance that the issues will be addressed. Wishing you the best of luck.

---

### Official Review · Reviewer_8xce · 2024-07-10

**Soundness:** 2
**Presentation:** 2
**Contribution:** 2
**Rating:** 6
**Confidence:** 4

**Summary:**

This paper studies the trade-off between in-distribution (ID) and out-of-distribution (OOD) performance of pre-trained models before and after fine-tuning. The authors observed that the sample distance is inversely proportional to $\frac{Acc_{ft}}{Acc_{zs}}$. After modeling the residual error of the model, they found that for a training sample $x$, the optimal ensemble weight is proportional to the sample's accuracy. Therefore, the authors set different ensemble weights for different samples based on the distance between the samples and the erroneous samples in the training set. Experimental results demonstrate the effectiveness of the proposed method.

**Strengths:**

1. This paper effectively improves the model's ID-OOD performance by setting different ensemble weights for different sample.
2. Experiments show that this method can be effectively applied to different fine-tuning techniques and can significantly enhance performance.

**Weaknesses:**

1. VRF requires identifying and saving the zs classification error samples for subsequent use, which presents certain limitations.
2. The description of Step 2 is unclear: is it calculating the distance to the k-th nearest sample in $V$, or is it clustering the representations in $V$ first and then calculating the distance to the $k$-th cluster center?
3. Is the representation $ v $ of $ x $ calculated by ft or zs?
4. For each sample, recalculating and sorting the distance to $V$ incurs additional inference costs. How much extra computational cost does this introduce?
5. Since line 278 indicates that different values of $k$ have a minor impact on the model, why not use the nearest sample to calculate the distance? What is the reason behind choosing the $k$-th sample for distance calculation?

**Questions:**

See Weaknesses.

**Limitations:**

Yes

---

> ### Author Rebuttal · Authors · 2024-08-06
>
> **Q1**: VRF requires identifying and saving the zs classification error samples for subsequent use, which presents certain limitations.
>
> **A1**: For our ImageNet experiments, the storage size for the ZSF set features is 289 MB. While this does require additional storage compared to some traditional methods, we believe it is a manageable overhead given the benefits in robustness and performance that our VRF framework provides.
>
> **Q2**: The description of Step 2 is unclear: is it calculating the distance to the k-th nearest sample in , or is it clustering the representations in  first and then calculating the distance to the -th cluster center?
>
> **A2**: We apologize for the unclear description. In Step 2, we calculate the distance to the k-th nearest sample in the ZSF set. We do not perform any clustering of the representations. We will clarify this in the revised version of the paper.
>
> **Q3**: Is the representation v of x calculated by ft or zs?
>
> **A3**: We use the representation from the fine-tuned model, as mentioned in Line 120: "we collect its feature representation from the fine-tuned model."’
>
> **Q4**: For each sample, recalculating and sorting the distance to incurs additional inference costs. How much extra computational cost does this introduce?
>
> **A4**: To address the efficiency of k-NN search, we leverage the Faiss library [11], which is optimized for large-scale similarity searches using GPUs. In our ImageNet experiments, the inference speed of the k-NN search for a single image is approximately 3.2 milliseconds, demonstrating that our approach can be executed efficiently even with large datasets.
>
> **Q5**: Since line 278 indicates that different values of  have a minor impact on the model, why not use the nearest sample to calculate the distance? What is the reason behind choosing the -th sample for distance calculation?
>
> **A5**: We avoid using the nearest sample to reduce the impact of label noise in the training set. If the nearest sample is mislabeled, the distance calculation could be unreliable. By selecting the $k$-th nearest sample, we mitigate this risk, as the likelihood of all $k$ nearest samples being mislabeled is low.

---

> > ### Comment · Reviewer_8xce · 2024-08-07
> >
> > I don't understand A5. The nearest sample is the case where $k=1$, Why does calculating $k$ using $float(p \cdot |V|)$ help mitigate the mislabeled noise? I think every sample has the same risk of being mislabeled, right?

---

> > > ### Author Response · Authors · 2024-08-08
> > > **Response to A5**
> > >
> > > You are correct in noting that each sample has an equal risk of being mislabeled. However, the density around the k-th sample is typically higher compared to the nearest sample. This higher density leads to lower variance when measuring distance using the k-th sample. Specifically, because the k-th sample is positioned between the (k-1)-th and (k+1)-th samples, in a high-density region, the range between these distances tends to be smaller. This reduced variance provides a more stable measure, making it less sensitive to outliers or mislabeled instances.
> > >
> > > Additionally, research[1] has demonstrated that using the k-th nearest distance is effective in density estimation in OOD detection, and we have adopted this approach.
> > >
> > > We hope this explanation addresses your concern.
> > >
> > > [1] Out-of-Distribution Detection with Deep Nearest Neighbors

---

> > > > ### Comment · Reviewer_8xce · 2024-08-08
> > > >
> > > > Thanks for your responese. I don't have any other questions and I have changed my score from 4 to 6.

---

> > > > > ### Author Response · Authors · 2024-08-09
> > > > >
> > > > > We do appreciate the reviewer's positive support. We will seriously take the reviewer's advice to carefully improve our paper. Thank you very much again.
> > > > >
> > > > > Sincerely, The authors of Submission 3218

---

### Official Review · Reviewer_e2mj · 2024-07-11

**Soundness:** 2
**Presentation:** 4
**Contribution:** 2
**Rating:** 5
**Confidence:** 4

**Summary:**

This paper aims to tackle the ID-OOD trade-off in the fine-tuning of pre-trained models with zero-shot abilities like CLIP. The proposed Variance Reduction Fine-tuning (VRF) is a sample-wise ensembling method concerning the zero-shot and fine-tuned models. The ensemble weights are determined by the distance from the test sample to the training samples that are incorrectly predicted by the zero-shot model. Theoretical analysis and experimental results justify the effectiveness of VRF.

**Strengths:**

1. The proposed VRF can achieve better performance on both ID and OOD data compared with existing ensemble-based methods.
2. Unlike previous methods, the ensemble weights in VRF does not require tuning.
3. The experiments are thorough, and the results are clearly presented in the tables and figures.

**Weaknesses:**

1. A major downside of VRF compared to previous ensemble methods is that the determination of the ensemble weights requires the storage of ZSF features at test time and computation of the distance from the test samples to each sample in ZSF set. This may bring a significant budget when the ZSF set is large. There should be a discussion on the theoretical and practical computational complexity regarding space and time.
2. The proposed distance calculation using the k-th nearest neighbor is borrowed from the previous work on OOD detection [24], and there is a lack of explanation for the specific choice of this metric. It seems that other metrics like the averaged k nearest neighbor distance discussed in [24] may also work in VRF.
3. It is claimed that the proposed method “can simultaneously attain the best ID and OOD accuracy without the trade-offs” (Line 9). However, this is not sufficiently justified. Specifically, it is unclear whether the “best ID and OOD accuracy” refers to the comparison with other existing methods or only considers the performance of the proposed method under different hyperparameter settings. Judging from Figure 3 (a.1), VRF does not always achieve the “best ID and OOD accuracy” compared with the peak ID/OOD accuracy of the existing ensembling method.

**Questions:**

1. To what extent is VRF less efficient than existing ensembling methods?
2. Why is k-th nearest neighbor distance selected for the distance calculation step of VRF?
3. What is the meaning of solving the ID-OOD trade-offs?

**Limitations:**

The authors have addressed some limitations and potential impacts of the work. I suggest additional discussions on the efficiency of the proposed method in the limitation part.

---

> ### Author Rebuttal · Authors · 2024-08-06
>
> **Q1**:  There should be a discussion on the theoretical and practical computational complexity regarding space and time.
>
> **A1**: To address the efficiency of k-NN search, we leverage the Faiss library [11], which is optimized for large-scale similarity searches using GPUs. In our ImageNet experiments, the inference speed of the k-NN search for a single image is approximately 3.2 milliseconds, demonstrating that our approach can be executed efficiently even with large datasets.
>
> For our ImageNet experiments, the storage size for the ZSF set features is 289 MB. While this does require additional storage compared to some traditional methods, we believe it is a manageable overhead given the benefits in robustness and performance that our VRF framework provides.
>
> We will include the discussion of the computational complexities in the revised version.
>
> **Q2**: The proposed distance calculation using the k-th nearest neighbor is borrowed from the previous work on OOD detection [24], and there is a lack of explanation for the specific choice of this metric. It seems that other metrics like the averaged k nearest neighbor distance discussed in [24] may also work in VRF.
>
> **A2**: We appreciate the reviewer’s observation and would like to clarify our rationale for choosing k-NN distance as the metric for density estimation in our VRF framework:
>
> * Training-Free and Easy Implementation: The k-NN method is training-free, making it simple to implement without requiring additional trained models. This aligns with our goal of maintaining a lightweight and efficient approach.
> * Non-Parametric Nature: k-NN is a non-parametric method, meaning it does not make any assumptions about the underlying distribution of the data. This flexibility contrasts with methods like those in [16, 22], which assume the feature space follows a Gaussian distribution, making k-NN a more robust choice across different datasets.
> * Strong ID and OOD Performance: We selected k-NN based on its demonstrated ability to effectively capture the relationship between distance measures and performance metrics (Acc_ft/Acc_zs) in both ID and OOD settings. This consistent performance across different scenarios reinforces our confidence in k-NN as a reliable metric.
>
> While we acknowledge that the averaged k-NN distance could also be a viable alternative, our experiments show that it yields similar results (82.3 ID and 61.8 OOD on ImageNet benchmarks) to the k-NN distance. However, the averaging step introduces additional computational complexity, albeit slight. To keep our method succinct and efficient, we chose to use k-NN distance.
> We will include the discussion in the revised version.
>
> **Q3**: It is unclear whether the “best ID and OOD accuracy” refers to the comparison with other existing methods or only considers the performance of the proposed method under different hyperparameter settings. Judging from Figure 3 (a.1), VRF does not always achieve the “best ID and OOD accuracy” compared with the peak ID/OOD accuracy of the existing ensembling method.
>
> **A3**: The statement “best ID and OOD accuracy” refers to the comparison with other existing methods. Our framework is designed to be orthogonal to existing ensembling techniques, meaning it can be applied in conjunction with them to enhance performance. Traditional ensembling methods often face trade-offs where optimizing for ID performance can degrade OOD performance, and vice versa. However, our VRF framework is intended to mitigate these trade-offs by simultaneously optimizing for both ID and OOD accuracy.
>
> In Figure 3 (a.1), the best OOD performance of conventional ensembling is 98.5%, while VRF achieves 98.4%. Although VRF is marginally lower by 0.1%, it is important to note that this difference is within a very small margin, particularly as both approaches are nearing the upper performance limit (close to 100%). Thus, we consider VRF's performance to be effectively on par with the best OOD accuracy, while also offering strong ID performance without requiring a trade-off.
>
> **Q4**: To what extent is VRF less efficient than existing ensembling methods?
>
> **A4**: The key distinction between VRF and conventional ensembling methods lies in their objectives. While traditional ensembling methods primarily focus on enhancing in-distribution (ID) performance, our VRF framework is designed to simultaneously improve both ID and out-of-distribution (OOD) performance. This broader objective naturally introduces additional computational overhead. However, as mentioned earlier, we utilize the Faiss library for efficient k-NN search, significantly reducing the time complexity. For example, in our ImageNet experiments, the k-NN search for a single image takes approximately 3.2 milliseconds, which we consider manageable given the overall performance benefits.
>
> **Q5**: why is k-th nearest neighbor distance selected.
>
> **A5**: We avoid using the nearest sample to reduce the impact of label noise in the training set. If the nearest sample is mislabeled, the distance calculation could be unreliable. By selecting the $k$-th nearest sample, we mitigate this risk, as the likelihood of all $k$ nearest samples being mislabeled is low.
>
>
> **Q6**: What is the meaning of solving the ID-OOD trade-offs?
>
> **A6**: Solving the ID-OOD trade-offs means developing models that can maintain high accuracy on in-distribution (ID) data while also performing well on out-of-distribution (OOD) data. Typically, improving OOD performance can reduce ID accuracy, but addressing these trade-offs aims to balance both, ensuring models are reliable in both familiar and new environments.

---

> > ### Author Response · Authors · 2024-08-12
> >
> > Dear Reviewer e2mj,
> >
> > We appreciate reviewer e2mj's valuable comments that significantly contribute to improving our manuscript.
> >
> > We want to know if the response address your concerns. Any further comments are welcome to us.
> >
> > Sincerely, The authors of Submission 3218

---

> > > ### Author Response · Authors · 2024-08-14
> > >
> > > Dear Reviewer e2mj,
> > >
> > > We sincerely appreciate your invaluable comments, which have significantly enhanced the quality of our manuscript. We hope our responses have adequately addressed your concerns. If you find our replies satisfactory, we kindly ask if you would consider revisiting your rating of our paper.
> > >
> > > Sincerely,
> > > The Authors

---

### Official Review · Reviewer_vpPs · 2024-07-13

**Soundness:** 4
**Presentation:** 3
**Contribution:** 3
**Rating:** 5
**Confidence:** 3

**Summary:**

This paper addresses the issue of ID-OOD (In-Distribution vs. Out-of-Distribution) in the context of robust fine-tuning techniques commonly used in ensemble methods. The authors propose a sample-wise mixing approach to resolve this problem. The method involves creating a Zero-Shot Failure set, which contains samples that fail in zero-shot models but succeed in fine-tuned models, and mixing the predictions of the zero-shot and fine-tuned models based on the distance between test samples and the Zero-Shot Failure set. This approach demonstrates superior performance over traditional weight-ensemble and prediction ensemble methods on the ImageNet variants benchmark.

**Strengths:**

- The paper is well-written, clearly explaining the proposed method and its motivation. The intuitive explanation, especially the use of Figure 2 to show the monotonic relationship between fine-tuned/zero-shot accuracy ratio and distance, strongly supports the proposed approach.
- The paper makes a novel contribution by addressing the ID-OOD problem, which is not fully addressed by the existing ensemble-based robust fine-tuning.
- The proposed method is simple and easy to implement, yet it achieves better performance compared to existing ensemble methods.

**Weaknesses:**

- The main drawback of the proposed method is the reliance on multiple hyperparameters (a, b, p), making tuning more complex compared to traditional ensemble methods that typically involve only one hyperparameter. Finding optimal values for these hyperparameters is challenging as they can vary significantly across different datasets, potentially affecting performance.
- To demonstrate the robustness of the proposed method, evaluations should be extended to more datasets beyond the ImageNet variants, similar to the experiments in the WiSE-FT paper, which include datasets like iWILDCam and FMoW. Differences in hyperparameter sweep ranges across datasets could highlight potential issues with the method.
- There is a lack of performance comparison with other robust fine-tuning methods, such as FLYP[1] and MaskFill[2], which limits the assessment of the proposed method’s effectiveness. In the ImageNet classificaiton experiments, the proposed method is inferior to [1,2] in performance.
- The paper need to report the hyperparameters found in experiments with different datasets and architectures to analyze their variability and sensitivity, providing deeper insights into the robustness of the proposed method.

[1] Finetune like you pretrain: Improved finetuning of zero-shot vision models, Goyal et. al., CVPR 23
[2] Masked images are counterfactual samples for robust fine-tuning, Xiao et. al., CVPR 23

**Questions:**

I am curious about the performance when finding hyperparameters for each test set as an oracle result.

**Limitations:**

The authors have adequately addressed some limitations of their work. However, they have not sufficiently discussed the critical issue of hyperparameter sensitivity and its impact on performance. There are no negative societal impacts.

---

> ### Author Rebuttal · Authors · 2024-08-06
>
> **Q1**:  The main drawback of the proposed method is the reliance on multiple hyperparameters.
>
> **A1**: We understand and acknowledge the reviewer’s concern regarding the complexity of tuning multiple hyperparameters. However, we would like to provide further clarification and context to address this issue:
> * No Need to Optimize p: First, we emphasize that the hyperparameter $p$ does not require optimization in our approach. We adopt the default value from [24], and as demonstrated in Figure 5(c), the performance is not sensitive to changes in $p$. To further validate this, we conducted experiments on CIFAR and Entity-30 datasets, where we varied $p$ from 0.0002% to 50%. The ID and OOD performance fluctuated by less than 0.1% and 0.3%, respectively, confirming that $p$ does not significantly impact the overall performance.
> * Simplified Search for a and b: Regarding the hyperparameters $a$ and $b$, the optimization process is straightforward and can be managed through grid search on the ID validation set. Additionally, we observed that the performance tends to peak at consistent values of $b$ even as $a$ varies. This pattern significantly reduces the search space for $b$, making the tuning process more efficient and less complex than it might initially appear.
>
> In conclusion, while our method involves multiple hyperparameters, we have shown that the tuning process is manageable and not as complex as it may seem. The robustness of $p$ and the structured tuning of $a$ and $b$ make our approach both practical and efficient across different datasets.
>
>
> **Q2**: Need to report the hyperparameters found in experiments with different datasets and architectures to analyze their variability and sensitivity.
>
> **A2**: We appreciate the reviewer’s insightful suggestion. We have reported the searched hyperparameters across different datasets and architectures below and will include them in the revised version of the paper.
>
> Our findings indicate that for the same dataset, the hyperparameters remain consistent across different architectures, suggesting that our method's performance is robust to architectural changes. However, the hyperparameters do vary between datasets, which is expected due to differences in data distribution. For example, the k-NN distance ranges from 0.4 to 1.6 for ImageNet and from 0.4 to 1.2 for Entity30, reflecting the underlying data characteristics. Correspondingly, the searched values for $a$ are 1.5 for ImageNet and 1.1 for Entity30. This variability in  $a$ highlights the importance of adapting hyperparameters to specific datasets to achieve optimal performance.
>
> ViT/32
>
> | Dataset | ImageNet | CIFAR | Entity30 |
> |---------|----------|-------|----------|
> | a       | 1.5      | 0.3   | 1.1      |
> | b       | 0.6      | 0.3   | 0.6      |
>
> ViT/16
>
> | Dataset | ImageNet | CIFAR | Entity30 |
> |---------|----------|-------|----------|
> | a       | 1.5      | 0.3   | 1.1      |
> | b       | 0.5      | 0.3   | 0.6      |
>
>
> **Q3**:  The performance when finding hyperparameters for each test set as an oracle result.
>
> We have conducted additional experiments where we optimized the hyperparameters for each test set of the ImageNet benckmarks. The results are summarized in the following table.
>
> | ViT/16     | IN   | IN-V2 | IN-S | IN-A | IN-R | ObjectNet |
> |------------|------|-------|------|------|------|-----------|
> | VRF        | 82.3 | 72.1  | 52.9 | 48.4 | 78.7 | 56.4      |
> | VRF oracle | 82.3 | 72.2  | 53.0 | 49.7 | 79.4 | 56.8      |
>
>
> **Q4**: iWILDCam and FMoW results.
>
> **Q4**: We have conducted experiments (E2E-FT model as the fine-tuned models) on fmow and iwildcam and report the performance with the hyper-parameters below. We will add the results of iWILDCam and FMoW in the revised version.
>
> | iWILDCam            | ID   | OOD  |
> |---------------------|------|------|
> | zero-shot           | 8.7  | 11.0 |
> | E2E-FT              | 48.0 | 34.7 |
> | +WSE                | 48.1 | 35.3 |
> | +OSE                | 48.0 | 35.0 |
> | +VRF (a=1.2, b=0.6) | 48.1 | 36.1 |
>
> | FMoW                | ID   | OOD  |
> |---------------------|------|------|
> | zero-shot           | 20.4 | 18.7 |
> | E2E-FT              | 68.5 | 39.2 |
> | +WSE                | 68.5 | 39.2 |
> | +OSE                | 68.5 | 39.2 |
> | +VRF (a=1.4, b=0.5) | 68.6 | 41.0 |
>
> **Q5**: Comparison with other robust fine-tuning methods.
>
> **A5**:  Our VRF framework is designed to be orthogonal and complementary to existing fine-tuned models. To demonstrate this, we conducted additional experiments using FLYP as the fine-tuned model within our VRF framework. The results show that our VRF framework enhances the performance of OSE by improving the distribution shift performance by 1.1% while maintaining the in-distribution (ID) performance.
>
>
> |      | IN   | V2   | R    | A    | S    | ObjectNet | OOD  |
> |------|------|------|------|------|------|-----------|------|
> | FLYP | 82.6 | 73.0 | 71.4 | 48.1 | 49.6 | 58.7      | 60.2 |
> | +WSE | 82.9 | 73.5 | 76.0 | 53.0 | 52.3 | 60.8      | 63.1 |
> | +OSE | 82.8 | 73.6 | 77.0 | 52.5 | 51.9 | 59.9      | 62.8 |
> | +VRF | 82.8 | 73.6 | 78.6 | 52.9 | 53.0 | 61.2      | 64.0 |

---

> > ### Author Response · Authors · 2024-08-12
> >
> > Dear Reviewer vpPs,
> >
> > We appreciate reviewer vpPs constructive feedback, which further helped us improve our draft.
> >
> > We have submitted our responses to concerns, and we want to know if these replies address your concerns. Any further comments or questions are welcome to us.
> >
> > Sincerely,
> >
> > The authors of Submission 3218

---

> > > ### Comment · Reviewer_vpPs · 2024-08-14
> > >
> > > Thank you for your detailed response, I have carefully reviewed all the reviews and the rebuttal content, and all of my concerns have been resolved.  In particular, after revisiting Figure 6, where the certain hyperparameter $b$ achieves a high peak in ID performance,has reassured me that the hyperparameters are searchable and generalizable to OOD datasets.
> > >
> > > To further strengthen your final paper, I would suggest including similar graphs for other datasets to confirm that these results are not unique to ImageNet. It would be good to show these in the appendix. Additionally, it would be valuable to provide graphs that illustrate the hyperparameters explored on the validation set, not just the test set's in-distribution performance.
> > >
> > > The proposed method effectively show the capability of sample-wise out-of-space ensemble, and thus, I would like to raise my score.

---

> > > > ### Author Response · Authors · 2024-08-14
> > > >
> > > > We are delighted that our rebuttal addresses your concerns and greatly appreciate your valuable suggestions. We also thank you for acknowledging the effectiveness of our method. We will include the graphs for all datasets, along with an exploration of the hyper-parameters on validation sets, in the revised manuscript.
> > > >
> > > > Sincerely,
> > > >
> > > > The authors of Submission 3218

---

### Author Rebuttal · Authors · 2024-08-06

We have uploaded ID-OOD frontier curves for WSE in the attachment.

---

### Decision · Program_Chairs · 2024-09-25

**Decision:**

Accept (poster)

**Comment:**

This paper presents an output-space ensemble approach to robust finetuning of zero-shot model (i.e., weighted sum of outputs from zero-shot and finetuned models). The reviewers appreciate the novel, simple, and effective method and its strong performance, and the clarity of the manuscript. They however at the same time raised multiple nontrivial concerns. Most reviewers were concerned about the high space-time complexity due to the double forward propagations and the k-th nearest neighbor distance computation (e2mj, 8xce, 52xu, ydTd). Other concerns include additional hyperparameters that may have to be carefully tuned per dataset (vpPs, 52xu), lack of comparisons with latest techniques (vpPs, ydTd), lack of empirical evidences supporting the versatility of the method (52xu), experiments on limited datasets (vpPs), design choices that are not well justified (e.g., the use of k-th nearest neighbor distance) (e2mj, 8xce), and some presentation issues (e2mj). The authors' rebuttal and subsequent responses in the discussion period address most of these concerns; a borderline reviewer was not fully satisfied by the response and still concerned in particular about the lack of sensitivity analysis, but mentioned that it is not a ground for rejection. In the end, all the reviewers were supportive of the paper.

The AC agrees with the reviewers and recommends acceptance. The authors are strongly encouraged to carefully revise the paper to reflect the valuable comments by the reviewers and to add new results and discussions brought up in the rebuttal and discussions, in particular a more thorough analysis on the sensitivity to hyperparameters and more experiments with a linear probed model; the latter is essential to fully resolve the concerns with the space-time complexity.